# Sharpness-Aware Minimization Scaled by Outlier Normalization for Improving Robustness on Noisy DNN Accelerators

## Abstract

Energy-efficient deep neural network (DNN) accelerators are prone to non-idealities that degrade DNN performance at inference time. To mitigate such degradation, existing methods typically add perturbations to the DNN weights during training to simulate inference on noisy hardware. However, this often requires knowledge about the target hardware and leads to a trade-off between DNN performance and robustness, decreasing the former to increase the latter. In this work, we first show that applying sharpness-aware training, by optimizing for both the loss value and loss sharpness, significantly improves robustness to noisy hardware at inference time without relying on any assumptions about the target hardware. Then, we propose a new adaptive sharpness-aware method that conditions the worst-case perturbation of a given weight not only on its magnitude but also on the range of the weight distribution. This is achieved by performing sharpness-aware minimization scaled by outlier minimization (SAMSON). Our extensive results on several models and datasets show that SAMSON increases model robustness to noisy weights without compromising generalization performance in noiseless regimes.

## 1 Introduction

The success of deep neural networks (DNNs) has been accompanied by an increase in training complexity and computational demands, prompting efficient DNN designs (Zhao et al., 2023; Lebovitz et al., 2023). With the slowing down of Moore's law and the ending of Dennard scaling, power consumption is now the key design constraint for DNN accelerators (Sze et al., 2020), which calls for new hardware and algorithms. In particular, in-memory computing approaches (Le Gallo et al., 2018; Sebastian et al., 2020; Yin et al., 2020; Sakr & Shanbhag, 2021) are promising directions to improve the energy consumption and throughput of existing DNNs by circumventing the need for memory accesses, which represent an energy-intensive process in conventional hardware implementations (Pedram et al., 2017). In-memory computing is especially important for running DNNs on edge devices which usually possess low-power constraints (Gupta et al., 2023).

Despite being highly energy-efficient, in-memory computing solutions require performing computations in the analog domain, which is inherently prone to variabilities (Xu et al., 2013). This leads to perturbations in the DNN weights after deploying it in the target hardware, ultimately resulting in a degradation in performance (Joshi et al., 2020; Kern et al., 2022; Spoon et al., 2021; Tambe et al., 2021). Hence, adopting such energy-efficient hardware designs requires developing DNN methods that improve robustness to hardware noise at inference time. The main approach for improving the hardware robustness of DNNs has been to apply weight perturbations during training (Hacene et al., 2019; Chang et al., 2019; Gokmen et al., 2019; Henwood et al., 2020; Joshi et al., 2020). However, these robustness methods provide a trade-off between DNN performance and DNN robustness, decreasing the former to increase the latter. Moreover, such approaches typically rely on information about the hardware device that the DNN will be deployed to (at inference time) to perform noise simulations at training time. However, this is not always possible and presents a limitation on the type of target hardware that can be used for inference.

The goal of this work is to increase model robustness without decreasing DNN performance and without relying on any noise simulations from the target hardware. By doing so, we do not compromise the applicability

of our approach, neither by reducing the original DNN performance nor by tailoring it to a specific hardware design. To achieve this, we propose a novel sharpness-aware minimization method that is applied during training to promote accurate DNN at inference time and after deployment on noisy, yet energy-efficient, hardware.

The benefit of converging to a smoother loss landscape has been primarily tied to improving generalization performance (Hochreiter & Schmidhuber, 1994; Keskar et al., 2016; Dziugaite & Roy, 2017; Neyshabur et al., 2017; Chaudhari et al., 2017; Izmailov et al., 2018). With this goal in mind, Foret et al. (2021) recently proposed sharpness-aware minimization (SAM) by minimizing both the loss value and loss sharpness within a maximization region around each parameter during training. By showing a high correlation between loss sharpness and test performance, SAM has ignited several follow-up works since its proposal. Particularly, adaptive SAM (ASAM) (Kwon et al., 2021) reformulated sharpness to be invariant to weight scaling by conditioning the neighborhood region of each weight based on its magnitude.

In this work, we propose to perform sharpness-aware minimization scaled by outlier normalization (SAM-SON) to increase robustness in DNNs without compromising performance. SAMSON reformulates adaptive sharpness to consider not only the weight magnitude but also the range of the weight distribution. By promoting sharpness adaptivity based on the outlier weights, we show that SAMSON's sharpness measure has a high correlation with model robustness. This suggests that the minimization of such sharpness measure by SAMSON's objective during training is an effective way to promote robustness after training, *i.e.* during inference. This is observed on a generic noise model on multiple DNN architectures and datasets as well as on accurate noise simulations from real hardware. Overall, our results showcase the extensive practicality of our approach by improving DNN robustness in noisy settings without affecting generalization performance in noiseless regimes.

## 2 Related work

The deployment of pre-trained models on noisy hardware for highly efficient inference is known to introduce non-idealities. This is caused by noise inherent to the device (Tsai et al., 2019) such as programming noise after weight transfer to the target hardware and read noise every time the programmed weights are accessed. Without robustness measures, such hardware noise significantly hinders the performance of neural networks. To promote robustness after deployment in noisy hardware at inference time, existing methods typically inject noise or faults to DNN weights during training (Ambrogio et al., 2018; Spoon et al., 2021; Li et al., 2019; Ambrogio et al., 2019; Mackin et al., 2019). In particular, adding weight noise (Joshi et al., 2020) and promoting redundancy by performing aggressive weight clipping (Stutz et al., 2021a; 2022) have been shown to be effective methods for increasing DNN robustness (Chitsaz et al., 2023). However, existing robustness methods often lead to a decrease of DNN performance for promoting robustness. Moreover, they typically rely on noise measurements from the target hardware to improve the performance and robustness trade-off. Here, we aim to increase DNN robustness in noisy settings without sacrificing DNN performance in the noiseless regime without relying on any information about the target hardware.

Sharpness-aware training has recently gathered increased interest (Sun et al., 2021; Jiang et al., 2020; Foret et al., 2021; Chen et al., 2022). Particularly, SAM has sparked a lot of new follow-up works due to the significant increase in generalization performance presented in the original paper. Variants mainly focus on increasing the efficiency (Du et al., 2022b;a; Zhou et al., 2022; Liu et al., 2022; Zhao et al., 2022), performance (Zhuang et al., 2022; Kim et al., 2022; Kwon et al., 2021), or understanding (Andriushchenko & Flammarion, 2022) of sharpness-aware training. Efforts have also been made to extend SAM to specific use-cases such as quantization-aware training (Liu et al., 2021b) or data imbalance settings Liu et al. (2021a). Several works (Kwon et al., 2021; Dinh et al., 2017) have also highlighted the importance of scale-invariant sharpness measures, including in the context of model robustness against adversarial examples (Stutz et al., 2021b).

In a similar vein to our work, Sun et al. (2021) recently related the sharpness of the loss landscape with robustness to adversarial noise perturbations. This was further observed by Kim et al. (2022). Stutz et al. (2021b) also recently studied the flatness of the (robust) loss landscape on the basis of adversarial training with perturbed examples (Madry et al., 2018). In particular, they tackle the problem of robust overfitting (He

et al., 2017), *i.e.* having high robustness to adversarial examples seen during training but generalizing poorly to new adversarial examples at test time, through the lens of flat minima. Our work, on the other hand, studies the relationship between the sharpness of the loss landscape and the robustness to hardware noise perturbations.

Particularly, our goal is to promote hardware robustness by retaining as much performance as possible while being exposed to hardware intrinsic variabilities that are reflected in terms of weight noise. This greatly differs from the malicious attacks performed in the context of adversarial robustness (Madry et al., 2018), even though both adversarial and hardware robustness share the common ground of degradation of the DNNs parameters. In the context of hardware robustness, however, the noise injected on the DNN weights depends on the variabilities of the target hardware to which the DNN is being deployed to. In other words, after training a DNN on reliable hardware, we study how deploying that pre-trained DNN to highly energy-efficient but noisy hardware would affect its performance at inference time.

## 3 Sharpness-aware minimization (SAM)

The goal of sharpness-aware minimization or SAM is to promote a smoother loss landscape by optimizing for both the loss value and loss sharpness during training. Generally speaking, given a parameter $w$, the goal is to find a region in the loss landscape where not only does $w$ have a low training loss $L$ but also do its neighbor points. Considering the $L_2$ norm and discarding the regularization term in the original algorithm for simplicity, SAM uses the following objective:

$$L_{\text{SAM}}(w) = \min_w \max_{\|\epsilon\|_2 \leq \rho} L(w + \epsilon), \tag{1}$$

where the size of the neighborhood region is defined by a sphere with radius $\rho$ and the optimal $\hat{\epsilon}$ may be efficiently estimated via a first-order approximation, leading to:

$$\boldsymbol{\epsilon}^*_{\text{SAM}}(\boldsymbol{w}) = \rho \, \frac{\nabla L(\boldsymbol{w})}{||\nabla L(\boldsymbol{w})||_2} \,. \tag{2}$$

By building on the strong correlation between sharpness and generalization performance, SAM is generally used in practice to achieve better test performance. However, there are two main drawbacks. The first is that, despite its efficiency in estimating the worst-case weight perturbations, SAM's update requires two backward passes. To mitigate this added complexity, the authors propose to leverage distributed training. Another drawback of SAM is that the sharpness calculation is not independent from weight scaling. This allows the manipulation of sharpness values by applying scaling operators to the weights such that weight values change without altering the model's final prediction (Dinh et al., 2017; Stutz et al., 2021b).

### 3.1 Adaptive SAM (ASAM)

To tackle the scale variance issue, adaptive sharpness-aware minimization or ASAM was proposed by Kwon et al. (2021). By taking into account scaling operators that do not change the model's loss, ASAM creates a new notion of adaptive sharpness that is invariant to parameter scaling, contrarily to SAM. This is reflected in ASAM's objective:

$$L_{\text{ASAM}}(w) = \min_w \max_{\|\epsilon/|w|\|_2 \leq \rho} L(w + \epsilon), \tag{3}$$

where $|w|$ represents the absolute value of a given weight $w$. With ASAM, different neighborhood sizes are applied to different weights, depending on their magnitude; high-magnitude weights withstand higher perturbations than low-magnitude weights. This adaptive sharpness formulation also leads to a change in the neighborhood shape, which is now ellipsoidal instead of spherical. The worst-case perturbation $\boldsymbol{\epsilon}^*_{\text{ASAM}}$ is defined as

$$\boldsymbol{\epsilon}^*_{\text{ASAM}}(\boldsymbol{w}) = \rho \, \frac{w^2 \nabla L(w)}{||w \nabla L(w)||_2} \,. \quad \text{(elementwise ops.)} \tag{4}$$

In practice, the adaptive sharpness that ASAM introduced shows a higher correlation with generalization performance and overall improved convergence by using larger maximization regions for larger weights.

# 4 Sharpness-aware minimization scaled by outlier normalization (SAMSON)

In this work, we propose a novel sharpness- and range-aware method called sharpness-aware minimization scaled by outlier normalization or SAMSON. In essence, our approach considers not only the weight magnitude but also the range of the weight distribution to determine the perturbation $\epsilon$ of a weight $w$. Conditioning sharpness by weight magnitude and the dynamic range of the weight distribution leads to the neighborhood sizes being normalized across all layers. This is particularly important when training with batch normalization, since the scales of the weight distributions across different layers may greatly differ leading to a discrepancy in the applied weight perturbations across the entire network.

We propose to take into account the outlier weight, *i.e.* the maximum absolute weight of a given layer, by simply scaling the effective neighborhood size of a weight $w$ by the $p$-norm of all the weights $\boldsymbol{w}$ in a given layer:

$$L_{\text{SAMSON}}(w) = \min_{w} \max_{\|\epsilon\|\boldsymbol{w}\|_p/|w|\|_2 \leq \rho} L(w + \epsilon), \tag{5}$$

which leads to the following per-weight worst-case perturbation:

$$\boldsymbol{\epsilon}^*_{\text{SAMSON}}(\boldsymbol{w}) = \rho \, \frac{(w\|\boldsymbol{w}\|_p^{-1})^2 \nabla L(w)}{\|w\|\boldsymbol{w}\|_p^{-1}\nabla L(w)\|_2} \, . \quad \text{(elementwise ops.)} \tag{6}$$

We note that the $p$-norm affects the impact of outlier weights in the applied worst-case perturbation. This differs from the norm ablations in Foret et al. (2021), where different norms are used to define the fixed (non-adaptive) neighborhood regions of all weights, with $\ell_2$-norm performing the best in practice. Without changing this default $\ell_2$-norm, our method uses different norms to control the importance of the outlier weights in the adaptive neighborhood region of each weight. In our study, we experiment with using $p = \{2, \infty\}$. For ease of presentation, we often refer to the variants with $p = 2$ and $p = \infty$ as SAMSON$_2$ and SAMSON$_\infty$, respectively, throughout the paper.

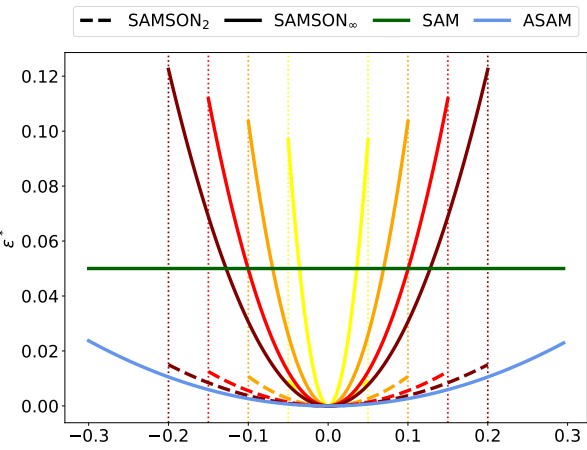

Figure 1: Worst-case perturbations of SAMSON, ASAM, and SAM. Each vertical, dotted line represents a different weight range [-$c$, $c$], with $c \in \{0.05, 0.1, 0.15, 0.2\}$.

We illustrate the applied worst-case perturbation considering a given weight value with SAMSON, ASAM, and ASAM in Fig. 1, assuming $\nabla L(w) = 1$ for simplicity. To showcase that our method is adaptive not only to the weight magnitude but also to the weight range, we apply different symmetric ranges to the original weight distribution. For the purpose of this illustration, we generate weight values between -0.3 and 0.3 using an interval of 0.005 and represent different weight clipping ranges ($\pm 0.2, \pm 0.15, \pm 0.1,$ and $\pm 0.05$) by vertical lines with different colors. We re-use the color to plot the effect that a given range has on the worst-case perturbations in SAMSON. We observe that $\epsilon^*_{\text{SAM}}$ is independent of the weight value and range, being represented as a straight line that is defined solely by $\rho$. Since $\epsilon^*_{\text{ASAM}}$ depends on $\rho$ and the

weight magnitude, larger weights are more perturbed. However, since ASAM is independent of the weight range, there is no change in ASAM's perturbations when changing the range of the weight distribution. On the other hand, SAMSON is both range- and weight magnitude-dependent, taking into account the weight value, $\rho$, and outlier weights for its perturbations. This results in the observed changes in $\epsilon^*_{\text{SAMSON}}$ over the different ranges, with SAMSON$_2$ putting less emphasis on outlier weights and SAMSON$_\infty$ emphasizing them.

Despite not depending on any form of weight clipping, SAMSON is inherently suited to be used in combination with methods that restrict the weight distribution range. For example, training with aggressive weight clipping (Stutz et al., 2021a) to improve robustness at inference time. When applying weight clipping, $c$ is the clipping range. With aggressive weight clipping, the weights are forced to be inside a small range to promote robustness: *i.e.* $c \in \mathbb{R} : 0 < c < 1$. A pseudo-code implementation of SAMSON combined with aggressive weight clipping is presented in Algorithm 1.

---

**Algorithm 1** SAMSON combined with weight clipping.

---

**Require:** initial weight $\boldsymbol{w_0}$, aggressive clipping range $c$, learning rate $\alpha$, neighborhood size $\rho$, norm $p$

$\quad \boldsymbol{w} \leftarrow \boldsymbol{w_0}$
$\quad$ **while** not converged **do**
$\quad\quad$ Sample minibatch $s$
$\quad\quad \boldsymbol{\epsilon} = \rho \frac{(w\|\boldsymbol{w}\|_p^{-1})^2 \nabla L_s(w)}{\|w\|\boldsymbol{w}\|_p^{-1}\nabla L_s(w)\|_2}$ $\qquad\qquad\qquad\qquad\qquad$ ▷ elementwise ops.
$\quad\quad \boldsymbol{w} \leftarrow \boldsymbol{w} - \alpha\nabla L_s(\boldsymbol{w}+\boldsymbol{\epsilon})$ $\qquad\qquad\qquad\qquad\qquad$ ▷ weight update
$\quad\quad \boldsymbol{w} \leftarrow \text{clip}(\boldsymbol{w}, c)$ $\qquad\qquad\qquad\qquad$ ▷ weight clipping (optional)
$\quad$ **end while**

---

## 5  Generalization performance

Before studying DNN robustness, we first analyze if using SAMSON negatively impacts generalization performance in the noiseless setting. Since this is the most common setting in practice, such a performance decrease would significantly reduce the applicability of our method. To test this, we first analyze the generalization performance of SAMSON, ASAM, SAM, and SGD on ResNet-34 (He et al., 2016), ResNet-50, MobileNetV2 (Sandler et al., 2018), VGG-13 (Simonyan & Zisserman, 2014), and DenseNet-40 (Huang et al., 2017) models trained on CIFAR-10 and CIFAR-100 (Krizhevsky & Hinton, 2009). All models were trained for 200 epochs with a batch size of 128, starting with a learning rate of 0.1 and dividing it by 10 every 50 epochs.

We used the default neighborhood sizes for SAM and ASAM, as proposed in their original papers: we set $\rho = 0.05$ and $\rho = 0.5$ for SAM and ASAM on CIFAR-10, respectively, and $\rho = 0.1$ and $\rho = 1.0$ for SAM and ASAM on CIFAR-100, respectively. For a direct method comparison, we report the results using the same default $\rho$ as ASAM for our method variants. When applicable, we report the mean and standard deviation over 3 runs. Additional details are presented in Appendix A.

The test accuracy comparisons between the different methods on CIFAR-10 and CIFAR-100 are shown in Tables 1 and 2, respectively. Overall, we observe that SAMSON does not lead to a decrease in generalization performance. and, in most cases, at least one of our variants (SAMSON$_2$ or SAMSON$_\infty$) even shows slight improvements over SGD, SAM, and ASAM in terms of test accuracy, with the best performing $p$ being dataset and architecture dependent. The only instance where a SAMSON variant is not the best-performing method is when using DenseNet-40 trained on CIFAR-10. However, both of our variants are within the standard deviation of ASAM, rendering the difference between the method performances statistically insignificant.

To further expand our exploration of models, datasets, and training settings, we finetuned a ResNet-18 model on ImageNet (Russakovsky et al., 2015) provided by PyTorch for a total of 10 epochs using SGD with momentum (0.9), a batch size of 400, a learning rate of 0.001, and a weight decay of 0.0001. Since no default $\rho$ is reported in the original ASAM's paper for finetuning on ImageNet, we iterate over different neighborhood ranges (details are provided in Appendix D) and report the best performing $\rho$ for SAM,

Table 1: Generalization performance (test accuracy %) of the different methods on several models trained on CIFAR-10.

| Method | ResNet-34 | ResNet-50 | MobileNetV2 | VGG-13 | DenseNet-40 |
|---|---|---|---|---|---|
| SGD | $95.84_{\pm 0.13}$ | $94.36_{\pm 0.09}$ | $94.62_{\pm 0.06}$ | $94.19_{\pm 0.04}$ | $91.76_{\pm 0.11}$ |
| SAM | $95.80_{\pm 0.07}$ | $94.24_{\pm 0.13}$ | $94.91_{\pm 0.07}$ | $94.52_{\pm 0.07}$ | $92.27_{\pm 0.30}$ |
| ASAM | $95.85_{\pm 0.22}$ | $94.42_{\pm 0.57}$ | $95.37_{\pm 0.04}$ | $94.68_{\pm 0.07}$ | $\mathbf{92.57_{\pm 0.06}}$ |
| SAMSON$_2$ | $\mathbf{95.96_{\pm 0.34}}$ | $\mathbf{95.09_{\pm 0.21}}$ | $95.29_{\pm 0.17}$ | $\mathbf{94.73_{\pm 0.12}}$ | $92.54_{\pm 0.14}$ |
| SAMSON$_\infty$ | $95.76_{\pm 0.29}$ | $\mathbf{94.94_{\pm 0.09}}$ | $\mathbf{95.41_{\pm 0.09}}$ | $94.66_{\pm 0.02}$ | $92.49_{\pm 0.13}$ |

Table 2: Generalization performance (test accuracy %) of the different methods on several models trained on CIFAR-100.

| Method | ResNet-34 | ResNet-50 | MobileNetV2 | VGG-13 | DenseNet-40 |
|---|---|---|---|---|---|
| SGD | $74.32_{\pm 1.32}$ | $74.35_{\pm 1.23}$ | $75.44_{\pm 0.07}$ | $72.78_{\pm 0.22}$ | $68.52_{\pm 0.25}$ |
| SAM | $75.62_{\pm 0.33}$ | $75.36_{\pm 0.01}$ | $76.81_{\pm 0.18}$ | $73.86_{\pm 0.40}$ | $69.14_{\pm 0.36}$ |
| ASAM | $76.91_{\pm 0.44}$ | $77.88_{\pm 0.85}$ | $77.28_{\pm 0.10}$ | $74.12_{\pm 0.01}$ | $70.21_{\pm 0.25}$ |
| SAMSON$_2$ | $\mathbf{77.68_{\pm 0.57}}$ | $\mathbf{78.22_{\pm 0.67}}$ | $77.24_{\pm 0.13}$ | $\mathbf{74.77_{\pm 0.23}}$ | $69.94_{\pm 0.36}$ |
| SAMSON$_\infty$ | $\mathbf{77.60_{\pm 0.78}}$ | $77.81_{\pm 1.32}$ | $\mathbf{77.61_{\pm 0.23}}$ | $74.59_{\pm 0.15}$ | $\mathbf{70.34_{\pm 0.37}}$ |

ASAM, and SAMSON. In the end, the best performances were obtained using $\rho = 0.05$ for SAM, $\rho = 0.2$ for SAMSON, and $\rho = 0.5$ for ASAM. Moreover, we also trained ResNet-18 and MobileNetV3 (Howard et al., 2019) models from scratch for 90 epochs using the same setup but with a learning rate of 0.1 decayed by 10 every 30 epochs. Results are presented in Table 3.

Table 3: Generalization performance (test accuracy %) of the different methods with ResNet-18 and MobileNetV3 on ImageNet.

| Method | Finetuned ResNet-18 | | Trained from scratch ResNet-18 | | MobileNetV3 | |
|---|---|---|---|---|---|---|
| | top-1 | top-5 | top-1 | top-5 | top-1 | top-5 |
| SGD | 69.758 | 89.078 | $69.91_{\pm .04}$ | $89.21_{\pm .05}$ | $69.30_{\pm .01}$ | $89.01_{\pm .01}$ |
| SAM | 70.356 | 89.480 | $70.01_{\pm .06}$ | $89.28_{\pm .06}$ | $69.32_{\pm .02}$ | $88.89_{\pm .02}$ |
| ASAM | 70.348 | 89.428 | $70.15_{\pm .06}$ | $89.24_{\pm .07}$ | $69.57_{\pm .08}$ | $88.90_{\pm .06}$ |
| SAMSON$_2$ | $\mathbf{70.358}$ | $\mathbf{89.486}$ | $70.16_{\pm .08}$ | $\mathbf{89.38_{\pm .10}}$ | $\mathbf{69.62_{\pm .01}}$ | $\mathbf{89.14_{\pm .01}}$ |
| SAMSON$_\infty$ | $\mathbf{70.366}$ | $\mathbf{89.504}$ | $\mathbf{70.23_{\pm .06}}$ | $89.35_{\pm .05}$ | $69.57_{\pm .03}$ | $88.99_{\pm .03}$ |

Once again, we observe that our variants do not degrade generalization performance, showing slight improvements over the compared methods when both fine-tuning or training from scratch. These results highlight the efficacy of our approach in achieving more robust DNNs (as will be discussed in the next sections) without degrading generalization performance in several training settings.

## 5.1 Large-scale Transformer models

To further assess SAMSON's ability to achieve competitive generalization performance in larger models, we present additional results using Transformers (Vaswani et al., 2017). Moreover, we also extend our covered tasks to include machine translation. As the Adam optimizer (Kingma & Ba, 2015) is the common practice in Transformers for text, we combined our method with Adam for the machine translation setup, further extending our analysis to applying SAMSON to additional optimizers on top of SGD.

For image classification, we finetuned a vision Transformer (ViT) model (Dosovitskiy et al., 2021), particularly a ViT-Base model (86.6M parameters) with $16 \times 16$ image patches, referred to as ViT-B-16, that was pre-trained on ImageNet-21K (Ridnik et al., 2021) and finetuned with SGD on ImageNet-1K resulting in a top-1 validation accuracy of 81.79%. In our experiments, we used the different sharpness-aware minimization methods to fine-tune this model for an additional 15 epochs. We used the same setup as previously described for ImageNet (including the previously presented $\rho$ for each method) except for the batch size which was set to 480 to maximize resource utilization. Results are presented in Table 4a. We observe that both of our method variants improve the generalization performance of the pre-trained model. On the other hand, ASAM was unable to recover the generalization performance of the original model, whereas SAM obtained a small improvement. Importantly, we see that both SAMSON variants outperform all the other methods.

For machine translation, we replicated the setup originally presented in ASAM (Kwon et al., 2021) and used an encoder-decoder, 12-layer Transformer (39.4M parameters) trained from scratch on IWSLT'14. Similarly to Kwon et al. (2021), we conducted a search over $\rho$ for our variants using the validation set (the range is presented in Appendix D), with $\rho = 0.5$ performing the best for both of our variants. For SAM and ASAM we used the best $\rho$ reported in Kwon et al. (2021). Results obtained over 3 seeds are shown in Table 4b. We observe that both of our method variants outperform all previous methods when evaluating the validation BLEU score. Moreover, SAMSON$_\infty$ outperforms all methods based on BLEU score using the test set. Overall, these results indicate that SAMSON can be successfully applied to achieve high generalization performance with large-scale Transformer models on image and language tasks.

Table 4: Generalization performance (validation accuracy % and validation and test BLEU scores) of the different methods with ViT-B-16 on finetuned ImageNet and an encoder-decoder Transformer trained on IWSLT'14. For IWSLT'14, the reported results (*) for Adam, Adam+SAM, and Adam+ASAM are taken from Kwon et al. (2021).

| Method | Finetuned ViT-B-16 val. accuracy |
|--------|----------------------------------|
| SGD | 81.79 |
| SAM | 81.80 |
| ASAM | 81.77 |
| SAMSON$_2$ | **81.83** |
| SAMSON$_\infty$ | **81.82** |

(a) ImageNet.

| Method | Trained from scratch Transformer | |
|--------|---------|----------|
| | val. BLEU | test BLEU |
| Adam | $35.34^*_{\pm<.01}$ | $34.86^*_{\pm<.01}$ |
| Adam+SAM | $35.52^*_{\pm.01}$ | $34.78^*_{\pm.01}$ |
| Adam+ASAM | $35.66^*_{\pm<.01}$ | $35.02^*_{\pm<.01}$ |
| Adam+SAMSON$_2$ | $\mathbf{35.70_{\pm<.01}}$ | $34.94_{\pm<.01}$ |
| Adam+SAMSON$_\infty$ | $\mathbf{35.73_{\pm<.01}}$ | $\mathbf{35.06_{\pm<.01}}$ |

(b) IWSLT'14.

# 6  Model robustness to noisy weights

We will now focus on analyzing how sharpness-aware training promotes DNN robustness compared to standard SGD training. In particular, we will focus on improving robustness in the context of noisy hardware accelerators that exploit the energy-reliability trade-off to improve energy efficiency at the cost of noisy weights. As our use-case, we consider memristor-based DNN implementations, which present a promising direction in energy-efficient DNN inference accelerators (Joshi et al., 2020; Kern et al., 2022). In such a setting, the weights of all fully-connected or convolutional layers of a pre-trained DNN are linearly mapped to the range of possible conductance values from 0 to $G_{\max}$. More concretely, the ideal conductance values $G^l_{T,ij}$ for the weights $W^l_{ij}$ of layer $l$ are

$$G^l_{T,ij} = \frac{W^l_{ij} \times G_{\max}}{W^l_{\max}} , \tag{7}$$

where $W^l_{\max}$ is layer $l$'s maximum absolute weight. However, as pointed out previously, $G^l_{T,ij}$ is not achievable in practice since conductance errors $\delta_{ij}$ are originated from programming and read noise (Tsai et al., 2019)

as well as conductance drift over time (Ambrogio et al., 2019). Hence, in the general case, the non-ideal conductance values $G_{ij}^l$ may be defined as

$$G_{ij}^l = G_{T,ij}^l \times \delta_{ij} \,, \tag{8}$$

with $\delta_{ij} \sim \mathcal{N}(1, \sigma_c^2)$. Following Joshi et al. (2020), $\sigma_c$ represents the conductance variation of $G_{ij}^l$ relative to $G_{T,ij}^l$. This generic noise model may be used to accurately estimate inference accuracy in noise models derived from measurements of existing noisy hardware implementations.

We tested robustness in a variety of networks – VGG-13 trained on CIFAR-10, MobileNetV2 trained on CIFAR-100, and ResNet-18 finetuned on ImageNet – following the same training procedure describe above. We tried a range of neighborhood sizes for the various methods since different a $\rho$ provides a distinct trade-off between performance and robustness. Additional details are provided in Appendix D. Overall, we found that $\rho = 0.5$ or $\rho = 1.0$ tend to provide the best trade-offs for both SAMSON and ASAM and $\rho = 0.05$ or $\rho = 0.1$ for SAM. To promote a cleaner visualization, we only report the best $\rho$ for each method. Lastly, we note that $\sigma_c = 0.0$ in our experiments refers to the special case where no noise is applied to the DNN weights.

## 6.1 Baseline robustness methods

On top of a simple baseline trained with vanilla SGD, we experimented with two methods: the additive noise approach proposed by Joshi et al. (2020) and aggressive weight clipping (Stutz et al., 2021a). More specifically, the first method applies additive Gaussian noise to DNN weights, whereas the second method clips the DNN weights into a small range of possible values. The models are trained from scratch and use the training settings previously described.

The additive random noise proposed by Joshi et al. (2020) is sampled from a Gaussian distribution $\mathcal{N}(0, \sigma_n^2)$, where

$$\sigma_n = \frac{W_{\max}^l \times \sigma_G}{G_{\max}} \,, \tag{9}$$

with $\sigma_G$ representing the standard deviation of hardware non-idealities observed in practice. Both $\sigma_G$ and $G_{\max}$ are device characteristics that are set to 0.94 and 25, respectively, following the empirical measurements on 1 million of phase-change memory devices (Joshi et al., 2020). Since the amount of added noise is proportional to the maximum absolute weight value of a given layer, we perform weight clipping after each weight update; we used the range $\left[-\alpha \times \sigma_{W^l}, \alpha \times \sigma_{W^l}\right]$, where $\sigma_{W^l}$ is the standard deviation of the weights of layer $l$ and $\alpha$ is a predefined hyper-parameter defaulted to 2.0. We tried a different range of $\alpha \in \{1.5, 2.0, 2.5\}$, but the best performance for all CIFAR-10/100 models was achieved with the default $\alpha$ value of 2.0. For finetuning on ImageNet, we used $\alpha = 2.5$, as originally suggested (Joshi et al., 2020).

For aggressive weight clipping, we tried the values for the clipping range $c$, as performed by the original authors (Stutz et al., 2021a): $\{\pm0.05, \pm0.10, \pm0.15, \pm0.20\}$. A lower weight range induced by a smaller $c$ leads to highly robust networks. However, they may lack generalization performance in the noiseless to low-noise regimes due to outlier distortion. Hence, manipulating $c$ provides a trade-off between performance and robustness. In our experiments, we observed that 0.2 (and in some cases 0.15) achieved the best trade-off and was used on most of the reported networks. Please see Appendix D for additional details.

To reduce the impact of hardware non-idealities in the DNN performance, Joshi et al. (2020) also proposed adaptive batch normalization statistics (AdaBS), which updates the batch normalization statistics using a calibration set. More specifically, the running mean and running variance of all batch normalization layers are updated using the statistics computed during inference on a calibration set using noisy weights. We used the originally suggested hyper-parameters and applied AdaBS to all networks.

## 6.2 Robustness to different conductance variation

The robustness of the models trained with SAMSON, ASAM, SAM, and SGD in combination with aggressive weight clipping at different conductance variation levels is shown in Fig. 2. We also include training with SGD and additive Gaussian noise as an additional baseline. For visualization clarity, we include training with additive noise on top of the sharpness-aware training variants in Appendix B.

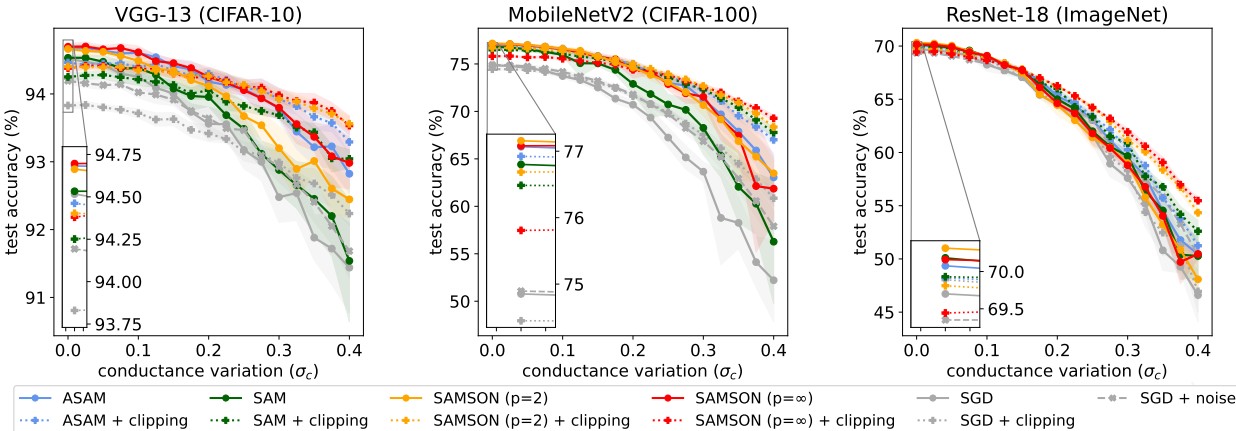

Figure 2: Performance of the different methods under a range of random conductance variations. We plot the mean and standard deviation over 10 and 3 inference runs for CIFAR-10/100 and ImageNet, respectively.

We observe that SAMSON variants primarily compose the Pareto frontier across all models and datasets. Ultimately, this means that training a DNN with SAMSON with and without aggressive weight clipping provides the best performance and robustness trade-off across all noisy regimes. This is also observed in the noiseless regime ($\sigma_c = 0.0$), where we see that there is always at least one SAMSON variant that achieves the best test accuracy, as discussed in section 5. The difference in model robustness between the various methods is more subtle on ImageNet, likely due to all methods starting with the same pre-trained model and being only finetuned for 10 epochs. Nevertheless, we see that SAMSON is the only method able to provide significant improvements in terms of robustness in highly noisy regimes, *e.g.* $\sigma_c = 0.4$.

Overall, we observe that sharpness-aware training variants (SAMSON, ASAM, and SAM) clearly outperform SGD, with SAMSON promoting the highest robustness, generally followed by ASAM and then SAM. This is seen in terms of not only robustness at different noise levels but also in the best performances achieved in the noiseless regime. Moreover, the improvement in robustness is especially amplified when combining sharpness-aware methods with aggressive weight clipping, representing a simple yet effective alternative to training with noise. We note that, as expected, the performance on the clean network drops when applying both weight clipping or additive noise, as observed in the zoomed-in patches. This mitigates the robustness benefits while using these methods in lower noisy settings but proves to be remarkably beneficial in highly noisy regimes.

### 6.2.1 Large-scale Transformer models

We further assessed the robustness of different conductance variations in our encoder-decoder Transformer model by applying noise to all the fully-connected layers. In this setting, we compare the different methods in combination with the Adam optimizer instead of SGD. The robustness results without aggressive weight clipping are presented in Fig. 3 (left). We observe that SAMSON$_\infty$ achieves the best robustness out of all the compared methods across all noisy regimes. Importantly, SAMSON$_2$ also achieves the second-best robustness in low to mid-range noisy settings, which are often the desirable noise ranges due to the maintenance of an adequate model performance for real-world tasks. Finally, in the zoomed-in patch, we see that both of our variants achieve the best generalization performance in the noiseless regime. (We note that we were unable to replicate the exact SAM and ASAM results reported in the original ASAM paper (Kwon et al., 2021) which explains the subtle difference in test BLEU scores presented here and Table 4b.)

We also experimented with applying aggressive weight clipping on this new model and task. Results are presented in Fig. 3 (right). In this setting, we observe that aggressive weight clipping did not help with increasing model robustness, but increased generalization performance. We hypothesize that this suggests that a smaller minimum and maximum clipping value may be used for this setting, instead of $c = \pm 0.2$ as

used in the reported results and also throughout the paper for image classification tasks. Notwithstanding, SAMSON$_2$ is the most viable robustness method when all the noise range is considered. Moreover, SAMSON$_\infty$ is also among the top-performing methods in the low to mid-noise regime. Lastly, we observe that once again both variants of our method achieve the best generalization performance in the noiseless setting, as presented in the zoomed-in patch.

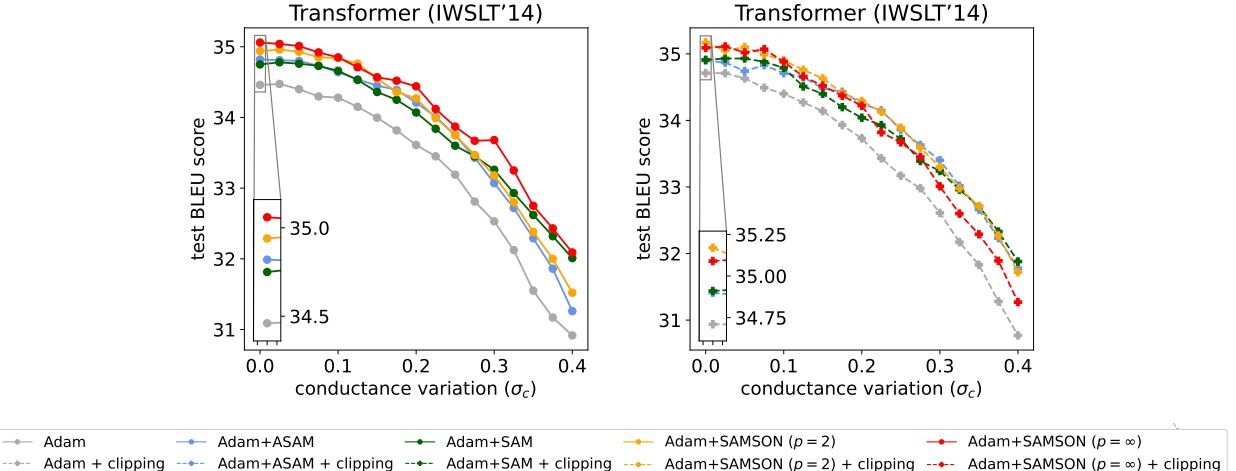

Figure 3: Performance of the different methods under a range of random conductance variations on Transformer with (right) and without (left) aggressive weight clipping. We plot the mean over 3 inference runs.

### 6.3 Sharpness and robustness correlation

For measuring sharpness, we use the $m$-sharpness metric proposed by Foret et al. (2021), which stems from the original SAM formulation (Eq. (1)), and further extend it to SAMSON's objective (Eq. (5)). Considering a training set ($S_{\text{train}}$) composed of $n$ minibatches $S$ of size $m$, we compute the difference of the loss $l_s$ of a given sample $s$ with and without a worst-case perturbation $\epsilon$ on $w$. SAMSON's $m$-sharpness is calculated as

$$\frac{1}{n} \sum_{S \in S_{\text{train}}} \max_{\|\epsilon\| \|\boldsymbol{w}\|_p^{-1}/|w|\|_2 \leq \rho} \frac{1}{m} \sum_{s \in S} l_s(w + \epsilon) - l_s(w). \tag{10}$$

In our experiments, we used $m = 400$ and $m = 128$ for measuring the sharpness of models finetuned on ImageNet and trained on CIFAR-10/100, respectively.

We treat robustness as the performance gap measured by the difference in test accuracy between the noiseless models, *i.e.* with no conductance variation applied to the weights ($\sigma_c = 0.0$), and the noisy model configurations with the highest tested conductance variation ($\sigma_c = 0.4$). We present the relation between sharpness and robustness of all the tested models using SAMSON's $m$-sharpness with $p = 2$ in Fig. 4. We observe a strong correlation within each training configuration, *i.e.* training each method with and without additive noise or aggressive weight clipping, across all architectures and datasets.

We provide visualizations of $m$-sharpness as calculated using SAMSON$_\infty$, SAM and ASAM's objectives and the metric proposed by Keskar et al. (2016) in Appendix E. We observe that SAMSON$_\infty$'s $m$-sharpness also shows a high correlation compared to the compared methods. Such findings showcase the ability of SAMSON's $m$-sharpness in acting as a generic robustness metric. Importantly, this suggests that training with SAMSON's objective, especially when combined with existing robustness methods such as aggressive weight clipping, is an effective way of promoting more robust DNNs at inference time.

### 6.4 Robustness to noise from real hardware

To convey how the performance on the generic noise model translates to existing hardware implementations, we performed experiments using an inference simulator on real hardware provided by IBM's analog hardware

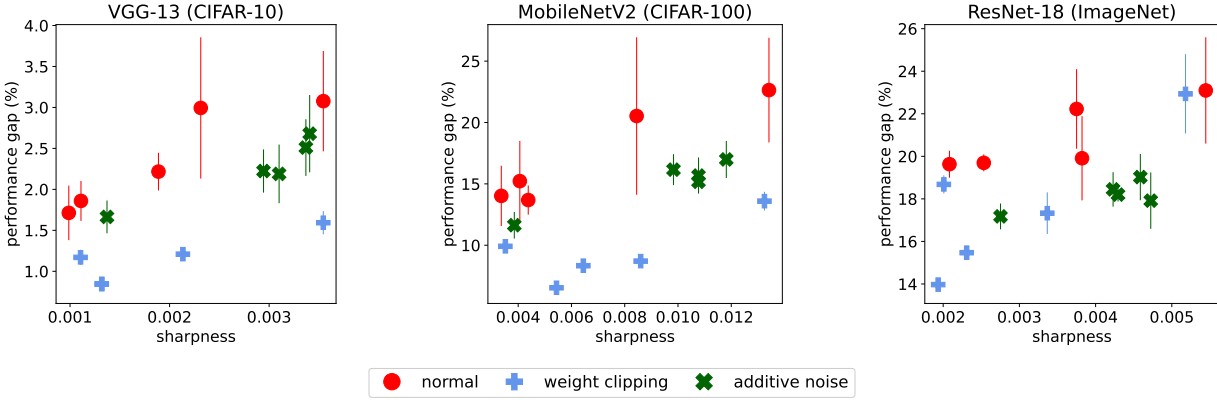

Figure 4: Correlation between SAMSON$_2$'s $m$-sharpness (Eq. (10), $\rho = 0.5$, $p = 2$) and robustness, *i.e.* the performance gap between the noise realizations at $\sigma_c = 0.0$ and at $\sigma_c = 0.4$. We plot the mean and standard deviation over 10 and 3 inference runs for CIFAR-10/100 and ImageNet, respectively.

acceleration kit (Rasch et al., 2021). This simulator uses the empirical measurements from 1 million phase-change memory devices (Nandakumar et al., 2019) to accurately simulate how hardware noise affects the DNN weights (Joshi et al., 2020). Specifically, by taking into account the programming and read noise, we report the performance of the different methods combined with aggressive weight clipping measured 1 year after deployment on the target hardware in Fig. 5. We observe that even though all sharpness-aware training methods outperform SGD in terms of robustness, the SAMSON variants retain the most performance. This is important in scenarios where often reprogramming the DNN weights on the memristor device is not feasible.

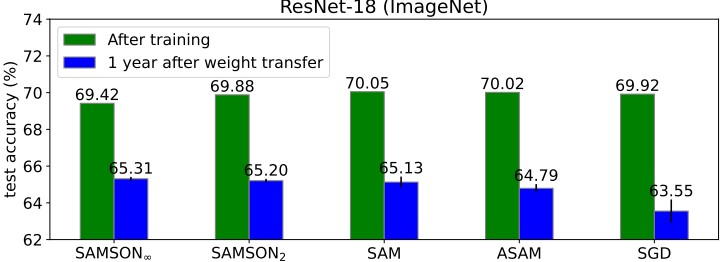

Figure 5: Performance of the different methods with aggressive weight clipping on ResNet-18 finetuned on ImageNet 1 year after weight transfer to the target hardware. We plot the mean and standard deviation over 10 inference runs.

## 7 Conclusion

In this work, we propose a new adaptive sharpness-aware training method that conditions the individual worst-case perturbation of a given weight based on not only its absolute value but also on the weight range distribution of a particular layer. Our results on different architectures, datasets, training regimes, and noisy scenarios showcase the benefits of using SAMSON to increase DNN robustness without compromising DNN performance in noiseless settings. One limitation of SAMSON which stems directly from SAM is the increase in training complexity. Notwithstanding, our approach may be combined with existing efficient SAM implementations (Du et al., 2022a; Liu et al., 2022) to further mitigate this issue.

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

## A  Training details

We trained the CIFAR-10/100 models using one RTX8000 NVIDIA GPU and 1 CPU core, the ResNet-18 models using one A100 40GB GPU and 6 CPU cores, and the Transformer models using one A100 80GB GPU and 6 CPU cores. For CIFAR-10/100, we used the architecture implementations in `https://github.com/kuangliu/pytorch-cifar`. For ImageNet, we used the ResNet-18 implementation provided by PyTorch [1]. For ViT-B-16 finetuned on ImageNet, we used the checkpoint publicly available at `https://huggingface.co/timm/vit_base_patch16_224.orig_in21k_ft_in1k`. For machine translation, we used the implementation in `https://github.com/facebookresearch/fairseq`.

## B  Additional robustness baselines

We also present the robustness results when combining the sharpness-aware training variants (SAM, ASAM, and SAMSON) with additive Gaussian noise in Fig. 6. Even though we observe an increase in robustness in certain configurations, training with aggressive weight clipping tends to provide the overall best trade-off between performance and robustness compared to training with additive noise.

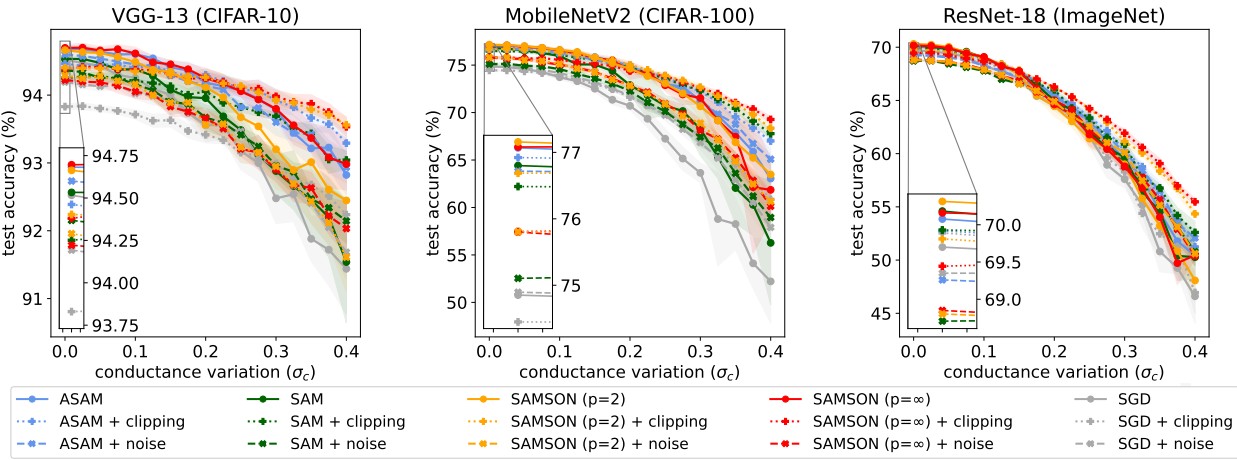

Figure 6: Performance of the different methods under a range of random conductance variations and combined with either weight clipping or additive noise. We plot the mean and standard deviation over 10 and 3 inference runs for CIFAR-10/100 and ImageNet, respectively.

## C  Additional robustness settings

To further expand the applicability of SAMSON, we also test its efficacy in promoting model robustness to out-of-distribution (OOD) examples and post-training quantization. Even though this falls outside the scope of hardware robustness that has been discussed so far, SAMSON is a generic robustness method that may be applied to different use cases such as robustness to OOD examples and quantization. We explore these additional robustness settings below.

### C.1  Out-of-distribution examples

To test robustness in OOD settings, we used 8 input perturbations as presented in Faramarzi et al. (2022), covering multiple scale, shear, and rotation transformations. We applied such perturbations to all pre-trained CIFAR-10 and CIFAR-100 models reported in Section 5, resulting in 40 scenarios per dataset. We report the number of times each method achieved the best test accuracy over 3 seeds and without any fine-tuning

---

[1]`https://pytorch.org/vision/main/models/generated/torchvision.models.resnet18.html`

Table 5: DenseNet-40 OOD experiments on CIFAR-10.

| Transform | SGD | SAM | ASAM | SAMSON$_2$ | SAMSON$_\infty$ |
|---|---|---|---|---|---|
| rotate$_{20}$ | $78.81_{\pm 0.62}$ | $78.38_{\pm 0.53}$ | $79.49_{\pm 0.93}$ | $79.25_{\pm 0.38}$ | $\mathbf{80.33_{\pm 0.82}}$ |
| rotate$_{40}$ | $56.86_{\pm 0.77}$ | $56.51_{\pm 0.69}$ | $57.51_{\pm 1.53}$ | $57.33_{\pm 1.73}$ | $\mathbf{58.73_{\pm 0.94}}$ |
| shear$_{28.6}$ | $79.47_{\pm 0.49}$ | $80.22_{\pm 0.69}$ | $81.41_{\pm 0.56}$ | $80.31_{\pm 0.85}$ | $\mathbf{81.65_{\pm 0.64}}$ |
| shear$_{57.3}$ | $54.49_{\pm 0.24}$ | $54.05_{\pm 0.69}$ | $56.08_{\pm 1.33}$ | $54.89_{\pm 1.82}$ | $\mathbf{56.69_{\pm 0.34}}$ |
| zoom$_{120}$ | $69.95_{\pm 1.88}$ | $70.08_{\pm 1.79}$ | $70.08_{\pm 1.65}$ | $\mathbf{71.49_{\pm 1.97}}$ | $70.64_{\pm 1.40}$ |
| zoom$_{140}$ | $39.18_{\pm 2.63}$ | $39.23_{\pm 3.93}$ | $38.65_{\pm 1.61}$ | $\mathbf{39.47_{\pm 0.34}}$ | $39.17_{\pm 1.13}$ |
| zoom$_{60}$ | $69.91_{\pm 0.53}$ | $71.26_{\pm 0.37}$ | $71.84_{\pm 1.43}$ | $\mathbf{72.72_{\pm 0.47}}$ | $72.60_{\pm 0.74}$ |
| zoom$_{80}$ | $85.53_{\pm 0.39}$ | $86.01_{\pm 0.27}$ | $86.48_{\pm 0.24}$ | $86.75_{\pm 0.39}$ | $\mathbf{87.08_{\pm 0.47}}$ |

in Fig. 7. We observe that our variants consistently outperform the existing methods on both datasets. Individual results for each OOD scenario in terms of test accuracies for CIFAR-10 and CIFAR-100 are shown in Tables 5 to 9 and Tables 10 to 14, respectively.

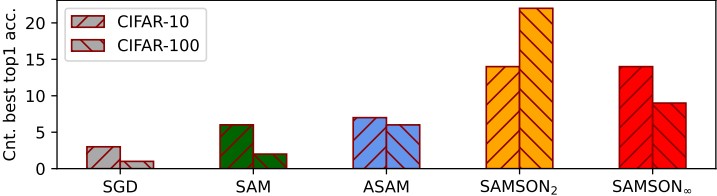

Figure 7: OOD robustness across 40 scenarios per dataset.

## C.2 Post-training quantization

To test robustness against post-training quantization (no fine-tuning), we used a linear quantization scheme without quantizing the first layer in all models. Results over 3 seeds using pre-trained MobileNetV2 and ResNet-18 models are presented in Fig. 8. While the robustness of our variants is similar to ASAM at medium to high bit-width, SAMSON$_\infty$ retains the most performance at the lowest bit-width for both models.

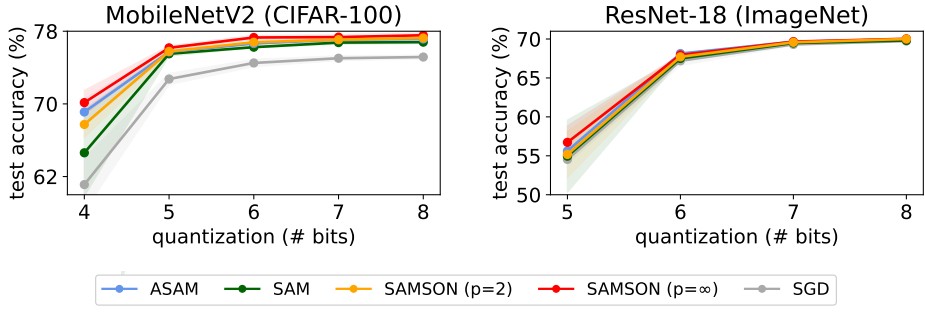

Figure 8: Robustness to quantization at different bit-widths.

## D Hyper-parameter tuning

The considered ranges for the different hyper-parameters are presented in Table 15. The configurations with the best performance and robustness trade-off for the models trained CIFAR-10, CIFAR-100, and ImageNet

Table 6: MobileNetV2 OOD experiments on CIFAR-10.

| Transform | SGD | SAM | ASAM | SAMSON$_2$ | SAMSON$_\infty$ |
|---|---|---|---|---|---|
| rotate$_{20}$ | $86.89_{\pm0.45}$ | $87.02_{\pm0.04}$ | $\mathbf{88.21_{\pm0.06}}$ | $87.47_{\pm0.42}$ | $87.43_{\pm0.48}$ |
| rotate$_{40}$ | $64.25_{\pm0.04}$ | $66.16_{\pm1.41}$ | $\mathbf{68.05_{\pm0.14}}$ | $66.24_{\pm0.04}$ | $67.17_{\pm0.37}$ |
| shear$_{28.6}$ | $85.49_{\pm0.04}$ | $86.82_{\pm0.18}$ | $\mathbf{87.70_{\pm0.16}}$ | $87.40_{\pm0.57}$ | $87.30_{\pm0.30}$ |
| shear$_{57.3}$ | $59.93_{\pm0.18}$ | $\mathbf{63.09_{\pm1.64}}$ | $62.56_{\pm1.51}$ | $62.10_{\pm0.64}$ | $62.56_{\pm0.70}$ |
| zoom$_{120}$ | $77.56_{\pm1.13}$ | $\mathbf{81.46_{\pm1.87}}$ | $79.80_{\pm1.02}$ | $80.97_{\pm1.41}$ | $80.97_{\pm1.35}$ |
| zoom$_{140}$ | $47.20_{\pm2.41}$ | $\mathbf{50.24_{\pm3.09}}$ | $49.28_{\pm0.18}$ | $48.62_{\pm2.03}$ | $48.62_{\pm1.92}$ |
| zoom$_{60}$ | $76.46_{\pm1.68}$ | $75.59_{\pm0.12}$ | $77.81_{\pm1.33}$ | $\mathbf{78.61_{\pm1.16}}$ | $\mathbf{78.61_{\pm0.30}}$ |
| zoom$_{80}$ | $90.11_{\pm0.45}$ | $90.69_{\pm0.22}$ | $\mathbf{91.53_{\pm0.21}}$ | $91.16_{\pm0.10}$ | $91.16_{\pm0.04}$ |

Table 7: ResNet-34 OOD experiments on CIFAR-10.

| Transform | SGD | SAM | ASAM | SAMSON$_2$ | SAMSON$_\infty$ |
|---|---|---|---|---|---|
| rotate$_{20}$ | $88.15_{\pm0.37}$ | $89.11_{\pm0.60}$ | $88.18_{\pm0.94}$ | $\mathbf{90.07_{\pm0.66}}$ | $89.29_{\pm0.58}$ |
| rotate$_{40}$ | $66.29_{\pm1.92}$ | $66.19_{\pm1.21}$ | $65.95_{\pm1.70}$ | $\mathbf{68.02_{\pm1.75}}$ | $66.81_{\pm0.40}$ |
| shear$_{28.6}$ | $86.76_{\pm0.57}$ | $88.46_{\pm0.58}$ | $86.88_{\pm0.94}$ | $\mathbf{89.05_{\pm1.19}}$ | $87.92_{\pm0.16}$ |
| shear$_{57.3}$ | $60.01_{\pm1.61}$ | $62.00_{\pm0.69}$ | $59.10_{\pm2.89}$ | $\mathbf{63.63_{\pm1.68}}$ | $62.12_{\pm0.60}$ |
| zoom$_{120}$ | $77.92_{\pm1.39}$ | $\mathbf{84.09_{\pm2.03}}$ | $78.98_{\pm2.07}$ | $77.14_{\pm1.23}$ | $80.40_{\pm0.61}$ |
| zoom$_{140}$ | $45.08_{\pm2.50}$ | $\mathbf{51.81_{\pm3.21}}$ | $46.74_{\pm2.57}$ | $44.22_{\pm1.59}$ | $48.41_{\pm3.74}$ |
| zoom$_{60}$ | $76.24_{\pm0.23}$ | $78.16_{\pm0.68}$ | $75.94_{\pm1.42}$ | $78.64_{\pm1.17}$ | $\mathbf{78.65_{\pm0.21}}$ |
| zoom$_{80}$ | $90.78_{\pm0.11}$ | $91.65_{\pm0.74}$ | $91.05_{\pm0.39}$ | $\mathbf{92.44_{\pm0.74}}$ | $92.09_{\pm0.43}$ |

Table 8: ResNet-50 OOD experiments on CIFAR-10.

| Transform | SGD | SAM | ASAM | SAMSON$_2$ | SAMSON$_\infty$ |
|---|---|---|---|---|---|
| rotate$_{20}$ | $85.46_{\pm0.06}$ | $84.79_{\pm0.52}$ | $84.40_{\pm1.60}$ | $\mathbf{86.30_{\pm0.94}}$ | $85.78_{\pm0.83}$ |
| rotate$_{40}$ | $\mathbf{65.45_{\pm1.47}}$ | $61.05_{\pm1.20}$ | $61.83_{\pm2.18}$ | $64.15_{\pm1.18}$ | $64.83_{\pm1.30}$ |
| shear$_{28.6}$ | $\mathbf{86.54_{\pm0.54}}$ | $84.95_{\pm0.47}$ | $84.67_{\pm1.06}$ | $86.40_{\pm0.80}$ | $86.44_{\pm0.58}$ |
| shear$_{57.3}$ | $61.63_{\pm1.20}$ | $58.37_{\pm1.94}$ | $59.97_{\pm0.90}$ | $\mathbf{62.33_{\pm0.54}}$ | $62.24_{\pm0.68}$ |
| zoom$_{120}$ | $77.05_{\pm4.03}$ | $70.94_{\pm1.78}$ | $74.80_{\pm1.47}$ | $77.17_{\pm2.81}$ | $\mathbf{80.09_{\pm4.54}}$ |
| zoom$_{140}$ | $\mathbf{51.10_{\pm7.88}}$ | $38.27_{\pm0.99}$ | $42.32_{\pm2.46}$ | $43.47_{\pm4.56}$ | $48.46_{\pm6.04}$ |
| zoom$_{60}$ | $73.57_{\pm0.81}$ | $74.70_{\pm0.57}$ | $70.48_{\pm2.99}$ | $74.92_{\pm1.37}$ | $\mathbf{76.30_{\pm0.23}}$ |
| zoom$_{80}$ | $88.94_{\pm0.28}$ | $89.29_{\pm0.28}$ | $88.84_{\pm0.74}$ | $89.62_{\pm0.80}$ | $\mathbf{90.56_{\pm0.14}}$ |

Table 9: VGG-13 OOD experiments on CIFAR-10.

| Transform | SGD | SAM | ASAM | SAMSON$_2$ | SAMSON$_\infty$ |
|---|---|---|---|---|---|
| rotate$_{20}$ | $85.00_{\pm0.69}$ | $86.56_{\pm0.28}$ | $86.81_{\pm0.08}$ | $86.81_{\pm0.21}$ | $\mathbf{87.19_{\pm0.36}}$ |
| rotate$_{40}$ | $64.46_{\pm1.71}$ | $67.05_{\pm0.52}$ | $\mathbf{67.06_{\pm0.90}}$ | $66.83_{\pm0.93}$ | $66.41_{\pm0.73}$ |
| shear$_{28.6}$ | $84.52_{\pm0.40}$ | $86.49_{\pm0.46}$ | $\mathbf{87.14_{\pm0.85}}$ | $86.66_{\pm0.32}$ | $86.61_{\pm0.34}$ |
| shear$_{57.3}$ | $58.56_{\pm0.43}$ | $59.18_{\pm0.36}$ | $\mathbf{61.14_{\pm1.25}}$ | $58.76_{\pm0.30}$ | $59.33_{\pm0.84}$ |
| zoom$_{120}$ | $84.12_{\pm0.09}$ | $85.73_{\pm1.46}$ | $84.58_{\pm0.69}$ | $\mathbf{86.46_{\pm1.48}}$ | $\mathbf{86.46_{\pm1.53}}$ |
| zoom$_{140}$ | $58.97_{\pm0.49}$ | $61.31_{\pm2.20}$ | $59.15_{\pm1.62}$ | $\mathbf{61.58_{\pm1.83}}$ | $\mathbf{61.58_{\pm2.35}}$ |
| zoom$_{60}$ | $72.11_{\pm0.61}$ | $\mathbf{74.34_{\pm2.18}}$ | $72.65_{\pm0.80}$ | $72.92_{\pm0.56}$ | $72.92_{\pm0.91}$ |
| zoom$_{80}$ | $88.37_{\pm0.37}$ | $89.66_{\pm0.03}$ | $89.66_{\pm0.35}$ | $\mathbf{90.06_{\pm0.23}}$ | $\mathbf{90.06_{\pm0.15}}$ |

Table 10: DenseNet-40 OOD experiments on CIFAR-100.

| Transform | SGD | SAM | ASAM | SAMSON$_2$ | SAMSON$_\infty$ |
|---|---|---|---|---|---|
| rotate$_{20}$ | $49.09_{\pm 0.29}$ | $49.74_{\pm 0.55}$ | $50.19_{\pm 0.52}$ | $\mathbf{50.51_{\pm 1.40}}$ | $50.11_{\pm 0.28}$ |
| rotate$_{40}$ | $31.89_{\pm 0.18}$ | $33.08_{\pm 0.27}$ | $32.74_{\pm 0.15}$ | $\mathbf{33.23_{\pm 1.46}}$ | $32.63_{\pm 0.84}$ |
| shear$_{28.6}$ | $52.59_{\pm 0.27}$ | $53.94_{\pm 0.35}$ | $54.19_{\pm 0.30}$ | $\mathbf{55.36_{\pm 0.98}}$ | $54.22_{\pm 0.53}$ |
| shear$_{57.3}$ | $34.60_{\pm 1.22}$ | $35.91_{\pm 0.26}$ | $36.00_{\pm 0.33}$ | $36.00_{\pm 1.03}$ | $\mathbf{36.50_{\pm 0.42}}$ |
| zoom$_{120}$ | $41.99_{\pm 1.71}$ | $41.52_{\pm 0.93}$ | $\mathbf{42.55_{\pm 1.90}}$ | $41.91_{\pm 0.66}$ | $42.11_{\pm 0.63}$ |
| zoom$_{140}$ | $\mathbf{16.93_{\pm 0.36}}$ | $15.24_{\pm 0.17}$ | $15.82_{\pm 0.70}$ | $15.82_{\pm 0.69}$ | $15.96_{\pm 1.04}$ |
| zoom$_{60}$ | $37.85_{\pm 2.06}$ | $38.46_{\pm 2.64}$ | $39.28_{\pm 1.06}$ | $\mathbf{40.67_{\pm 0.68}}$ | $39.24_{\pm 0.91}$ |
| zoom$_{80}$ | $58.49_{\pm 0.55}$ | $59.35_{\pm 0.22}$ | $60.01_{\pm 0.14}$ | $\mathbf{60.66_{\pm 0.37}}$ | $60.40_{\pm 0.40}$ |

Table 11: MobileNetV2 OOD experiments on CIFAR-100.

| Transform | SGD | SAM | ASAM | SAMSON$_2$ | SAMSON$_\infty$ |
|---|---|---|---|---|---|
| rotate$_{20}$ | $62.28_{\pm 0.63}$ | $62.76_{\pm 0.08}$ | $63.47_{\pm 0.40}$ | $\mathbf{64.59_{\pm 0.52}}$ | $64.09_{\pm 0.32}$ |
| rotate$_{40}$ | $42.97_{\pm 0.36}$ | $42.95_{\pm 0.18}$ | $44.09_{\pm 0.46}$ | $\mathbf{44.31_{\pm 0.11}}$ | $44.28_{\pm 0.37}$ |
| shear$_{28.6}$ | $61.50_{\pm 0.43}$ | $64.42_{\pm 0.59}$ | $64.88_{\pm 0.29}$ | $\mathbf{65.01_{\pm 0.37}}$ | $64.72_{\pm 0.09}$ |
| shear$_{57.3}$ | $42.13_{\pm 0.88}$ | $42.80_{\pm 1.70}$ | $42.62_{\pm 0.48}$ | $\mathbf{43.89_{\pm 0.76}}$ | $42.98_{\pm 0.09}$ |
| zoom$_{120}$ | $54.60_{\pm 0.99}$ | $53.32_{\pm 3.61}$ | $\mathbf{60.53_{\pm 1.21}}$ | $56.40_{\pm 1.53}$ | $55.13_{\pm 2.88}$ |
| zoom$_{140}$ | $26.69_{\pm 1.36}$ | $24.88_{\pm 2.40}$ | $\mathbf{34.06_{\pm 2.05}}$ | $27.29_{\pm 2.28}$ | $27.46_{\pm 2.83}$ |
| zoom$_{60}$ | $45.51_{\pm 2.21}$ | $43.48_{\pm 0.59}$ | $42.58_{\pm 2.23}$ | $42.89_{\pm 2.65}$ | $\mathbf{46.33_{\pm 0.23}}$ |
| zoom$_{80}$ | $66.62_{\pm 0.36}$ | $67.85_{\pm 0.18}$ | $69.30_{\pm 0.39}$ | $68.93_{\pm 0.55}$ | $\mathbf{69.65_{\pm 0.29}}$ |

Table 12: ResNet-34 OOD experiments on CIFAR-100.

| Transform | SGD | SAM | ASAM | SAMSON$_2$ | SAMSON$_\infty$ |
|---|---|---|---|---|---|
| rotate$_{20}$ | $62.62_{\pm 0.63}$ | $64.40_{\pm 0.22}$ | $65.34_{\pm 0.45}$ | $65.17_{\pm 0.50}$ | $\mathbf{66.01_{\pm 0.57}}$ |
| rotate$_{40}$ | $43.59_{\pm 0.14}$ | $44.03_{\pm 1.09}$ | $45.49_{\pm 0.24}$ | $45.32_{\pm 0.30}$ | $\mathbf{45.77_{\pm 0.64}}$ |
| shear$_{28.6}$ | $62.63_{\pm 0.70}$ | $65.03_{\pm 0.53}$ | $65.44_{\pm 1.03}$ | $66.06_{\pm 0.29}$ | $\mathbf{66.20_{\pm 0.38}}$ |
| shear$_{57.3}$ | $40.27_{\pm 1.15}$ | $40.81_{\pm 0.74}$ | $43.63_{\pm 0.57}$ | $\mathbf{44.79_{\pm 0.59}}$ | $43.56_{\pm 0.39}$ |
| zoom$_{120}$ | $59.78_{\pm 0.92}$ | $61.12_{\pm 2.42}$ | $62.42_{\pm 0.44}$ | $\mathbf{64.59_{\pm 2.41}}$ | $60.50_{\pm 1.19}$ |
| zoom$_{140}$ | $37.58_{\pm 2.85}$ | $37.17_{\pm 3.68}$ | $38.50_{\pm 0.80}$ | $\mathbf{42.03_{\pm 4.63}}$ | $35.37_{\pm 2.76}$ |
| zoom$_{60}$ | $43.14_{\pm 1.07}$ | $45.32_{\pm 0.97}$ | $46.27_{\pm 0.42}$ | $\mathbf{47.24_{\pm 2.00}}$ | $46.00_{\pm 1.62}$ |
| zoom$_{80}$ | $66.09_{\pm 0.71}$ | $68.34_{\pm 0.67}$ | $\mathbf{70.22_{\pm 0.56}}$ | $70.10_{\pm 0.84}$ | $69.57_{\pm 0.35}$ |

Table 13: ResNet-50 OOD experiments on CIFAR-100.

| Transform | SGD | SAM | ASAM | SAMSON$_2$ | SAMSON$_\infty$ |
|---|---|---|---|---|---|
| rotate$_{20}$ | $59.44_{\pm 0.80}$ | $60.36_{\pm 0.70}$ | $60.37_{\pm 1.46}$ | $\mathbf{64.53_{\pm 1.58}}$ | $61.84_{\pm 0.57}$ |
| rotate$_{40}$ | $40.10_{\pm 1.53}$ | $40.47_{\pm 1.21}$ | $40.45_{\pm 1.61}$ | $\mathbf{44.73_{\pm 1.44}}$ | $42.80_{\pm 0.50}$ |
| shear$_{28.6}$ | $59.70_{\pm 1.87}$ | $62.47_{\pm 0.68}$ | $62.71_{\pm 1.01}$ | $\mathbf{65.63_{\pm 0.27}}$ | $63.30_{\pm 1.05}$ |
| shear$_{57.3}$ | $38.96_{\pm 1.05}$ | $40.56_{\pm 0.79}$ | $42.45_{\pm 0.27}$ | $\mathbf{45.57_{\pm 1.82}}$ | $41.57_{\pm 1.72}$ |
| zoom$_{120}$ | $53.62_{\pm 2.09}$ | $57.13_{\pm 1.49}$ | $54.54_{\pm 5.03}$ | $\mathbf{57.97_{\pm 0.00}}$ | $57.05_{\pm 1.07}$ |
| zoom$_{140}$ | $27.24_{\pm 3.47}$ | $\mathbf{29.51_{\pm 0.75}}$ | $26.04_{\pm 6.95}$ | $29.35_{\pm 0.95}$ | $29.21_{\pm 0.86}$ |
| zoom$_{60}$ | $40.36_{\pm 3.12}$ | $45.47_{\pm 1.60}$ | $45.19_{\pm 2.65}$ | $\mathbf{47.96_{\pm 1.16}}$ | $44.71_{\pm 1.99}$ |
| zoom$_{80}$ | $63.45_{\pm 2.57}$ | $66.28_{\pm 0.32}$ | $67.63_{\pm 1.65}$ | $\mathbf{69.77_{\pm 0.33}}$ | $67.78_{\pm 1.91}$ |

Table 14: VGG-13 OOD experiments on CIFAR-100.

| Transform | SGD | SAM | ASAM | SAMSON$_2$ | SAMSON$_\infty$ |
|---|---|---|---|---|---|
| rotate$_{20}$ | $56.80_{\pm 0.45}$ | $58.52_{\pm 0.86}$ | $59.77_{\pm 0.36}$ | $\mathbf{59.87_{\pm 0.30}}$ | $59.85_{\pm 0.43}$ |
| rotate$_{40}$ | $38.53_{\pm 0.71}$ | $40.14_{\pm 0.25}$ | $\mathbf{40.68_{\pm 0.40}}$ | $40.45_{\pm 0.43}$ | $40.30_{\pm 0.47}$ |
| shear$_{28.6}$ | $58.88_{\pm 0.08}$ | $60.45_{\pm 0.45}$ | $60.68_{\pm 0.15}$ | $\mathbf{60.97_{\pm 0.48}}$ | $60.70_{\pm 0.21}$ |
| shear$_{57.3}$ | $36.70_{\pm 0.24}$ | $38.53_{\pm 0.47}$ | $38.29_{\pm 0.39}$ | $38.58_{\pm 0.27}$ | $\mathbf{39.85_{\pm 1.23}}$ |
| zoom$_{120}$ | $56.94_{\pm 0.76}$ | $60.30_{\pm 1.63}$ | $\mathbf{61.03_{\pm 0.12}}$ | $59.71_{\pm 0.49}$ | $59.06_{\pm 1.36}$ |
| zoom$_{140}$ | $33.18_{\pm 1.00}$ | $\mathbf{37.00_{\pm 2.70}}$ | $35.98_{\pm 1.00}$ | $35.09_{\pm 0.23}$ | $34.04_{\pm 1.31}$ |
| zoom$_{60}$ | $35.52_{\pm 0.42}$ | $38.30_{\pm 0.15}$ | $39.10_{\pm 1.14}$ | $40.72_{\pm 0.71}$ | $\mathbf{41.33_{\pm 1.23}}$ |
| zoom$_{80}$ | $64.11_{\pm 0.85}$ | $65.33_{\pm 0.47}$ | $66.28_{\pm 0.14}$ | $66.03_{\pm 0.12}$ | $\mathbf{66.48_{\pm 0.30}}$ |

Table 15: Hyper-parameter choices for the different methods.

| Hyper-parameter | Choices |
|---|---|
| SAM's $\rho$ | $\{0.05,\ 0.1,\ 0.2,\ 0.5\}$ |
| ASAM's $\rho$ | $\{0.5,\ 1.0,\ 1.5,\ 2.0\}$ |
| SAMSON's $p$ | $\{2,\ \infty\}$ |
| SAMSON's $\rho$ | $\{0.1,\ 0.2,\ 0.5,\ 1.0\}$ |
| $c$ | $\{\pm 0.05,\ \pm 0.10,\ \pm 0.15,\ \pm 0.20\}$ |
| $\alpha$ | $\{1.5,\ 2.0,\ 2.5\}$ |

are presented in tables 16, 17, and 18, respectively. These configurations were the ones used to report the results in the main paper.

# E  Additional sharpness experiments

We also provide correlation results with additional sharpness metrics. Particularly, we analyze the $m$-sharpness as formulated per SAM and ASAM's objectives. For SAM, $m$-sharpness is calculated as

$$\frac{1}{n} \sum_{S \in S_{\text{train}}} \max_{\|\epsilon\|_2 \leq \rho} \frac{1}{m} \sum_{s \in S} l_s(w + \epsilon) - l_s(w), \tag{11}$$

whereas for ASAM, $m$-sharpness is obtained by

$$\frac{1}{n} \sum_{S \in S_{\text{train}}} \max_{\|\epsilon/|w|\|_2 \leq \rho} \frac{1}{m} \sum_{s \in S} l_s(w + \epsilon) - l_s(w). \tag{12}$$

To avoid repetition, we refer to the main paper for notations. Visual correlations between loss sharpness and model robustness using SAMSON$_\infty$, SAM, and ASAM's $m$-sharpness are presented in figs. 9, 10, and 11, respectively. Results using Keskar et al. (2016)'s sharpness are also shown in Fig. 12. Overall, we see that SAMSON$_\infty$ shows the highest visual correlation, comparatively with the SAMSON$_2$ shown in the main paper. Moreover, we observe that both SAM's and ASAM's $m$-sharpness show better visual correlation than Keskar et al. (2016)'s notion of sharpness. This suggests that optimizing for low sharpness during training by using existing sharpness-aware training methods is an effective way to promote robustness at inference time, as discussed in the main paper.

Table 16: Best hyper-parameter configurations for VGG-13 trained on CIFAR-10.

| Method | Best configuration |
|---|---|
| SGD + noise | $\alpha = 2.0$ |
| SGD + clipping | $c = \pm 0.15$ |
| SAM | $\rho = 0.1$ |
| SAM + noise | $\rho = 0.1, \alpha = 2.0$ |
| SAM + clipping | $\rho = 0.1, c = \pm 0.2$ |
| ASAM | $\rho = 0.5$ |
| ASAM + noise | $\rho = 0.5, \alpha = 2.0$ |
| ASAM + clipping | $\rho = 0.5, c = \pm 0.2$ |
| SAMSON$_2$ | $\rho = 0.2, p = 2$ |
| SAMSON$_2$ + clipping | $\rho = 0.5, p = 2, c = \pm 0.2$ |
| SAMSON$_2$ + noise | $\rho = 0.1, p = 2, \alpha = 2.0$ |
| SAMSON$_\infty$ | $\rho = 1.0, p = \infty$ |
| SAMSON$_\infty$ + clipping | $\rho = 0.5, p = \infty, c = \pm 0.2$ |
| SAMSON$_\infty$ + noise | $\rho = 0.1, p = \infty, \alpha = 2.0$ |

Table 17: Best hyper-parameter configurations for MobileNetV2 trained on CIFAR-100.

| Method | Best configuration |
|---|---|
| SGD + noise | $\alpha = 2.0$ |
| SGD + clipping | $c = \pm 0.2$ |
| SAM | $\rho = 0.2$ |
| SAM + noise | $\rho = 0.2, \alpha = 2.0$ |
| SAM + clipping | $\rho = 0.2, c = \pm 0.2$ |
| ASAM | $\rho = 1.0$ |
| ASAM + noise | $\rho = 1.0, \alpha = 2.0$ |
| ASAM + clipping | $\rho = 1.0, c = \pm 0.2$ |
| SAMSON$_2$ | $\rho = 1.0, p = 2$ |
| SAMSON$_2$ + clipping | $\rho = 0.5, p = 2, c = \pm 0.2$ |
| SAMSON$_2$ + noise | $\rho = 0.2, p = 2, \alpha = 2.0$ |
| SAMSON$_\infty$ | $\rho = 1.0, p = \infty$ |
| SAMSON$_\infty$ + clipping | $\rho = 1.0, p = \infty, c = \pm 0.2$ |
| SAMSON$_\infty$ + noise | $\rho = 0.2, p = \infty, \alpha = 2.0$ |

Table 18: Best hyper-parameter configurations for ResNet-18 finetuned on ImageNet.

| Method | Best configuration |
|---|---|
| SGD + noise | $\alpha = 2.5$ |
| SGD + clipping | $c = \pm 0.2$ |
| SAM | $\rho = 0.1$ |
| SAM + noise | $\rho = 0.05, \alpha = 2.5$ |
| SAM + clipping | $\rho = 0.1, c = \pm 0.2$ |
| ASAM | $\rho = 1.0$ |
| ASAM + noise | $\rho = 0.5, \alpha = 2.5$ |
| ASAM + clipping | $\rho = 1.0, c = \pm 0.2$ |
| SAMSON$_2$ | $\rho = 0.2, p = 2$ |
| SAMSON$_2$ + clipping | $\rho = 0.5, p = 2, c = \pm 0.2$ |
| SAMSON$_2$ + noise | $\rho = 0.1, p = 2, \alpha = 2.5$ |
| SAMSON$_\infty$ | $\rho = 0.5, p = \infty$ |
| SAMSON$_\infty$ + clipping | $\rho = 1.0, p = \infty, c = \pm 0.2$ |
| SAMSON$_\infty$ + noise | $\rho = 0.1, p = \infty, \alpha = 2.5$ |

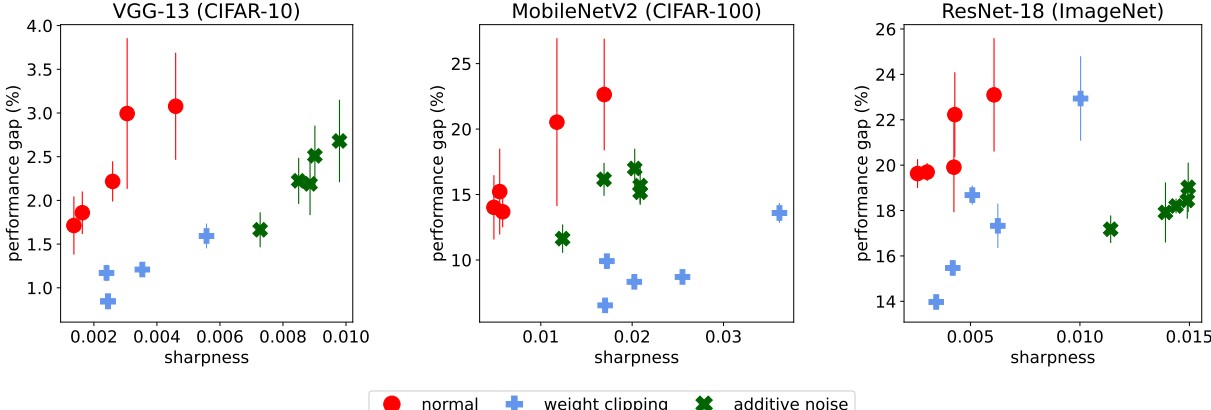

Figure 9: Correlation between SAMSON$_\infty$'s $m$-sharpness (Eq. (10), $\rho = 0.5$, $p = \infty$) and robustness, *i.e.* the performance gap between the noise realizations at $\sigma_c = 0.0$ and at $\sigma_c = 0.4$. We plot the mean and standard deviation over 10 and 3 inference runs for CIFAR-10/100 and ImageNet, respectively.

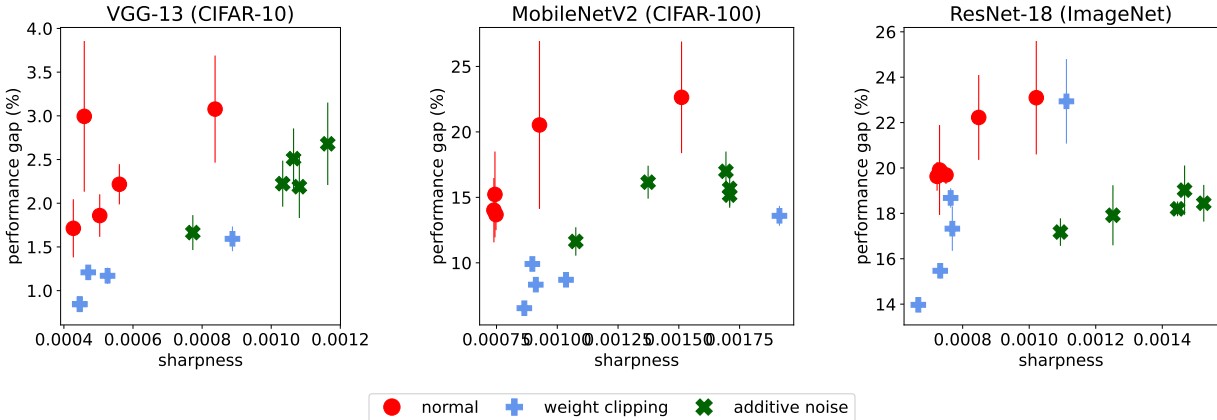

Figure 10: Correlation between SAM's $m$-sharpness (Eq. (11), $\rho = 0.05$) and robustness, *i.e.* the performance gap between the noise realizations at $\sigma_c = 0.0$ and at $\sigma_c = 0.4$. We plot the mean and standard deviation over 10 inference runs.

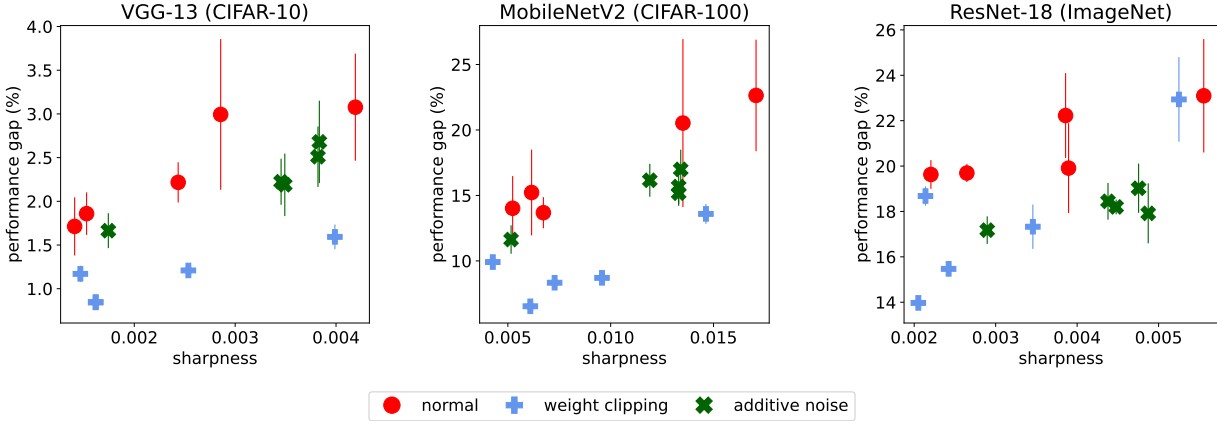

Figure 11: Correlation between ASAM's $m$-sharpness (Eq. (12), $\rho = 0.5$) and robustness, *i.e.* the performance gap between the noise realizations at $\sigma_c = 0.0$ and at $\sigma_c = 0.4$. We plot the mean and standard deviation over 10 inference runs.

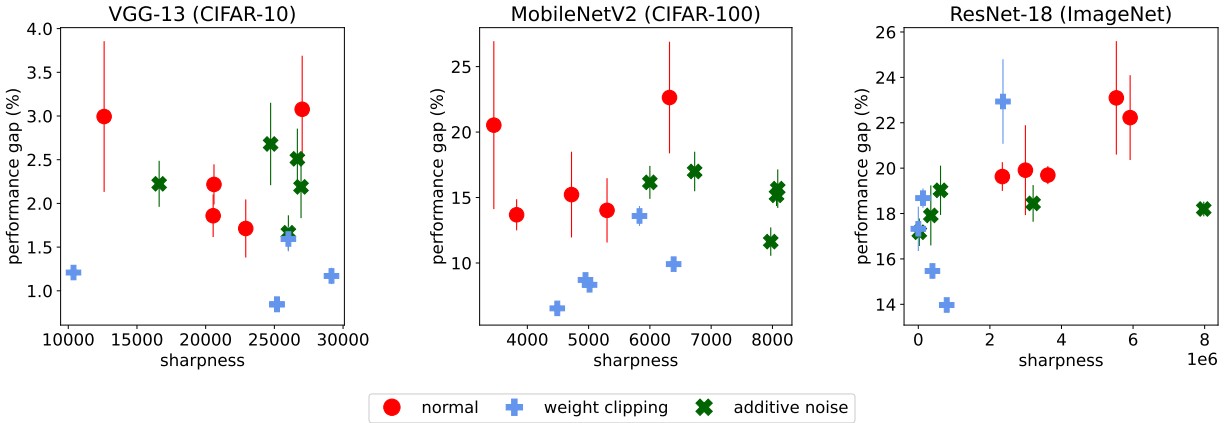

Figure 12: Correlation between Keskar et al. (2016)'s sharpness and robustness, *i.e.* the performance gap between the noise realizations at $\sigma_c = 0.0$ and at $\sigma_c = 0.4$. We plot the mean and standard deviation over 10 inference runs.

### E.1 Pearson correlations

We present the Pearson correlations and p-values of the different $m$-sharpness methods and generalization gaps in Table 19. Overall, we observe a high Pearson coefficient and small p-values, suggesting a statistically significant correlation between the sharpness-aware minimization objectives of the different methods and robustness, in particular with our variants.

Table 19: Pearson coefficients and p-values between different sharpness measures and robustness on different models, datasets, and training regimes: normal (N), additive noise (AN), and weight clipping (WC).

| $m$-sharpness | CIFAR-10 VGG-13 | | | CIFAR-100 MobileNetV2 | | | ImageNet ResNet-18 | | |
|---|---|---|---|---|---|---|---|---|---|
| | N | AN | WC | N | AN | WC | N | AN | WC |
| SAM | $.59_{p=.28}$ | $.93_{p=.02}$ | $.89_{p=.04}$ | $.87_{p=.05}$ | $.78_{p=.11}$ | $\mathbf{.90_{p=.03}}$ | $\mathbf{.94_{p=.01}}$ | $.88_{p=.04}$ | $\mathbf{.94_{p=.01}}$ |
| ASAM | $\mathbf{.92_{p=.02}}$ | $.93_{p=.01}$ | $.87_{p=.05}$ | $\mathbf{.98_{p<.01}}$ | $.92_{p=.02}$ | $.73_{p=.15}$ | $.85_{p=.06}$ | $.76_{p=.13}$ | $.84_{p=.07}$ |
| SAMSON$_2$ | $\mathbf{.92_{p=.02}}$ | $.93_{p=.01}$ | $.86_{p=.05}$ | $.95_{p=.01}$ | $\mathbf{.95_{p=.01}}$ | $.72_{p=.16}$ | $.84_{p=.07}$ | $.76_{p=.13}$ | $.84_{p=.07}$ |
| SAMSON$_\infty$ | $.91_{p=.02}$ | $\mathbf{.96_{p<.01}}$ | $\mathbf{.89_{p=.03}}$ | $.97_{p<.01}$ | $.80_{p=.09}$ | $.83_{p=.07}$ | $.86_{p=.05}$ | $\mathbf{.91_{p=.02}}$ | $\mathbf{.94_{p=.01}}$ |

## F Additional discussions

As we propose a simple extension to the ASAM method, we seek to propose an adaptive sharpness that is invariant to parameter scaling across the considered parameter tensor. With respect to ASAM, which is invariant to parameter scaling on a per parameter basis, our approach defines a neighbourhood region in the loss landscape that is influenced by the statistics of the parameters tensor. It encourages a high $\epsilon$ for small weights if the weight tensor it belongs to has values in the same range, whereas ASAM would simply not apply much perturbation to it. One scenario where the range of the weight tensor is small and the weight elements share close values is when using aggressive weight clipping to promote better robustness. With aggressive weight clipping, *i.e.* $c \in \mathbb{R} : 0 < c < 1$, $\epsilon^*_{\text{SAMSON}_\infty}(w)$ is strictly bigger than $\epsilon^*_{\text{ASAM}}(w)$ at the same $\rho$:

$$\rho \frac{(wc^{-1})^2 \nabla L(w)}{||wc^{-1}\nabla L(w)||_2} > \rho \frac{w^2 \nabla L(w)}{||w\nabla L(w)||_2} \tag{13}$$

$$\frac{1}{\sqrt{c^2}} > 1. \tag{14}$$

However, achieving a good trade-off between low loss and low sharpness is inherent to any sharpness-aware minimization method due to the minmax nature of the objective. Hence, we note that controlling the amount of perturbations may be beneficial to achieve a better trade-off at the end of training. In our approach, this can be done by using $p = 2$, which softens SAMSON's worst-case perturbation when compared to $p = \infty$. Since the loss-sharpness trade-off is task and model-specific, it is recommended in practice to treat the $p$-norm as a hyperparameter to account for different training dynamics.

Another scenario is when the weight ranges of different layers are not in similar intervals due to the use of batch normalization for example. To illustrate, let us define a 2-layer DNN with parameters $w_1$ and $w_2$. We further suppose that throughout the training of the DNN, the parameters in $w_1$ observe a large magnitude, and those of $w_2$ have a small magnitude, e.g. close to zero, with respect to the average values of both tensors: that is $||w_1||_2 >> ||w_2||_2$. In ASAM's formulation, the perturbation applied to $w_2$ would be close to zero, whereas $w_1$ would have large perturbations, i.e. $||\epsilon^*_{\text{ASAM}}(w_1)|| >> ||\epsilon^*_{\text{ASAM}}(w_2)||$. SAMSON aim to put the perturbation on the same scale for $w_1$ and $w_2$ by scaling the neighbourhood to account for the statistics of $w_1$ and $w_2$, that is $||\epsilon^*_{\text{SAMSON}}(w_1)|| \approx ||\epsilon^*_{\text{SAMSON}}(w_2)||$. In other words, SAMSON aims to close a gap in the definition of ASAM by automatically tuning the neighborhood size depending on the parameter tensor statistics.

Considering our use-case of DNN deployment on memristor devices, where parameters are scaled on a per tensor basis, the range of the tensor defines the maximum relative impact of the hardware noise. The tighter the range, the less of an effect the hardware noise will have. SAMSON reflects this behaviour by scaling $\epsilon$ with respect to the range and as such is more related to such hardware setting than ASAM or SAM.

### F.1 Limitations

A limitation of our method is regarding the types of sharpness considered. Particularly, SAMSON's objective optimizes for the $m$-sharpness measure presented in Eq. (10). Such measure depends on the worst-case perturbation within the neighborhood size $\rho$ which which can be characterized as minimizing for zeroth-order flatness. Such flatness has been recently shown by Zhang et al. (2023) to achieve a lower maximum eigenvalue of the Hessian and Hessian trace compared to the first-order flatness proposed by the previous work. This leads to minima that may have sharp curvature in certain directions instead of low curvature across all directions. Combining our approach with such first-order flatness solution is an interesting future work.

