# OpenReview forum: "Sharpness-Aware Minimization Scaled by Outlier Normalization for Improving Robustness on Noisy DNN Accelerators"
_TMLR — Rejected by TMLR_

### Review · Reviewer_V41i · 2023-11-08

**Summary Of Contributions:**

The paper proposes SAMSON, an extension of adaptive Sharpness-Aware Minimization (ASAM). The main difference is that the L_p-norm of the overall weight tensor is taken into account when determining the neighborhood size of a weight w used in SAM. This is hypothesized to increase correlation of the resulting sharpness measure with robustness to noisy weights and thus help when deploying a DNN for in-memory computing solutions that require performing computations in the analog domain. Benefits of SAMSON over baselines is demonstrated on different image classification benchmarks.

**Audience:**

Yes

**Broader Impact Concerns:**

No broader impact concerns

**Claims And Evidence:**

Yes

**Requested Changes:**

- Additional experiments on stronger models and/or more diverse type of tasks would strengthen the evidence for the claims of the paper.
- The correlation between sharpness and robustness in Figure 3 does not appear to be strong; it should be also quantified (e.g. Pearson's correlation coefficient) and tested for statistical significance
- From the description, it appears that the L_p norm is computed over the entire weight tensor. Is this correct? Because an alternative would have been to compute it on a per-layer basis. This would also be a reasonable additional ablation experiment.

Minor: The formatting of Table 1 is unfortunate (in the middle of a page, would be better at top or bottom of page
Minor: Text in Section 7.1 states "fine-tuning in Section 7.1", where it should probably be "in Figure 5"

**Strengths And Weaknesses:**

Strengths:
 * the paper is in general well written and easy to follow
 * relevant related work is summarized and the proposed method is compared to relevant prior work
 * the proposed SAMSON is well motivated and a reasonable (albeit simple) extension of Adaptive SAM
 * SAMSON is compared to a reasonable set of baselines in terms of in-distribution generalization, robustness to noisy weights (with realistic noise model from real hardware), out-of-distribution generalization, and post-training quantization. Also, combinations with weight clipping are investigated. Overall, SAMSOM performs favourably.

Weaknesses:
 * The experiments are performed on a relatively narrow setting, focusing only on image classification and excluding modern architectures such as Vision Transformers.
* Obtained performance on test sets is relatively far from state-of-the-art methods, peaking below 71% accuracy on ImageNet. It is unclear how well the proposed method would scale to higher performing models

In summary, the paper is well written and presents a reasonable but small change to Adaptive SAM. It would benefit from stronger empirical evidence of the method's utility in more state-of-the-art and more diverse settings.

---

> ### Author Response · Authors · 2023-12-01
>
> Thank you for your valuable comments. We provide detailed answers to your requested changes below.
>
> > “Additional experiments on stronger models and/or more diverse type of tasks would strengthen the evidence for the claims of the paper.”
>
> Thank you for this suggestion. We present additional large-scale experiments using an encoder-decoder Transformer ($39.4$M parameters) for machine translation and a vision Transformer ($86.6$M parameters) for image classification. The results are presented in **Table 4** of **Section 5.1**. Overall, we observe that our method can successfully be applied to improve the generalization performance of stronger models on diverse types of tasks (i.e. image classification and machine translation). We also tested robustness to noisy weights for the Transformer model used in machine translation in **Figure 3** of **Section 6.2.1**. We observe that our method is a viable approach to promote robustness in this new setting.
>
> > “The correlation between sharpness and robustness in Figure 3 does not appear to be strong; it should be also quantified (e.g. Pearson's correlation coefficient) and tested for statistical significance”
>
> We agree that having an objective measure of the correlation between sharpness and robustness is a great addition to the paper. We present Pearson's correlation coefficients and the p-values for the different $m$-sharpness methods in **Table 19** of **Appendix E.1**. Overall, we observe a high and statistically significant correlation between SAMSON’s $m$-sharpness and model robustness.
>
> > “From the description, it appears that the L_p norm is computed over the entire weight tensor. Is this correct? Because an alternative would have been to compute it on a per-layer basis. This would also be a reasonable additional ablation experiment.”
>
> Thank you for pointing this out. We would like to clarify that we compute the $p$-norm on a per-layer basis and have now made this clear in **Section 4** by changing the text before introducing Eq. (5).

---

> > ### Comment · Reviewer_V41i · 2023-12-11
> > **Reviewer Response**
> >
> > I would like to thank the authors for their feedback. The revised paper addresses my concerns and is in a good shape now.

---

### Review · Reviewer_GL5H · 2023-11-11

**Summary Of Contributions:**

This work shows that applying sharpness-aware training significantly improves robustness to noisy hardware at inference time without relying on any assumptions about the target hardware. It proposes a new adaptive sharpness-aware method that conditions the worst-case perturbation of a given weight not only on its magnitude but also on the range of the weight distribution. Results on several models and datasets in terms of robustness to noisy weights, out-of-distribution examples, and post- training quantization show that SAMSON increases model robustness. The reviewer holds a concern that the title is a bit over-claiming in terms of "DNN robustness" and the introduction might confuses the audience about the scope and contribution.

**Audience:**

Yes

**Broader Impact Concerns:**

No concern.

**Claims And Evidence:**

No

**Requested Changes:**

High-level:
1. The introduction only focuses on efficient DNN designs in terms of the energy consumption, throughput, DNN accelerators etc.. However, it is necessary to introduce the motivation about the robustness problem you will be focusing at the beginning instead of talking only about the energy consumption in the introduction which seems to be unrelated to your work. Could you discuss at least some prior works on improving robustness (e.g., [1,2]) in the introduction?
2. I only realized this paper only focus on "hardware robustness" which, according to the definition in the related work, indicates the robustness to the noise caused by different hardware initialization. I am very confused when reading the first two paragraphs about what is exactly the scope this paper is studying. For example, the paper mentioned OOD and quantization extension later which is unrelated to the hardware robustness. I believe when you say "robustness" in the title, it should include more broad scope like distribution shift and spurious correlation. In addition, you said Sun et al. (2021)'s work on adversarial robustness is related to your work, which from the hardware robustness definition, it isn't related right? I don't understand what is the meaning of saying "Even though we also study robustness to out-of-distribution examples and quantization, we mainly focus on the problem of improving robustness against noisy weights at inference time in this work." at the end of related work. Please clarify what is actual problem investigated in this work and how is it related to all other related works.
3. Table 1 & 2 shows the accuracy comparisons between the different methods on CIFAR-10 and CIFAR-100. I think they are just showing the vanilla accuracy, but why the captions say they show the "Generalization performance" as no noise is added during evaluation?


Low-level:
1. Could you explain a bit more about how the first two paragraphs in the introduction sections relate to the main contribution of this paper? I don't quite understand why the energy consumption and in-memory computation for DNNs can be a motivation of this robustness study, e.g., if you want to claim the traditional adversarial training [1] is inefficient, you might want to cite it properly instead of only talking about general DNN accelerator design without the context of adversarial training. Besides, would be great to also mention the efficient versions of adversarial training, e.g., [2], and discuss their limitations that your method can address.
2. In the second paragraph in the introduction section, it says "However, such approaches typically rely on noise simulations from the target hardware to which the DNN will be deployed.". I'm confused about the definition of the "noise simulations from the target hardware", probably you can move its definition from related work to introduction.
3. Please provide citations to "existing methods" when you say "Moreover, existing robustness methods provide a trade-off between DNN performance and DNN robustness, decreasing the former to increase the latter." in the introduction section.
4. If adversarial robustness is the scope, please discuss related works on SAM+adversarial robustness evaluation, e.g., [3]. If not, please clarify this point and explain why Sun et al. (2021)[4]'s work on adversarial robustness is the most related.
5. In the last paragraph in the introduction section, you say "By pro- moting sharpness adaptivity based on the outlier weights, we show that SAMSON’s sharpness measure has a high correlation with model robustness. In other words, SAMSON’s objective may be used during training in combination with existing robustness techniques to increase DNN robustness at inference time.". However, I don't think the two sentences have the same meaning, right?

[1] Madry et al.,"Towards Deep Learning Models Resistant to Adversarial Attacks".\
[2] Shafahi et al., "Adversarial Training for Free!".\
[3] Shao et al., "On the Adversarial Robustness of Vision Transformers".\
[4] Sun et al., "Exploring the vulnerability of deep neural networks: A study of parameter corruption."\

**Strengths And Weaknesses:**

Strengths:
This paper proposed a new modification on ASAM, which helps improve the hardware robustness of DNNs and can be combined with other SOTA methods to improve the efficiency.

Weekness:
The title is over-claiming. The paper only discuss about hardware robustness in the main text, i.e., the robustness against noise caused by hardware initialization etc.. However, its title says the method is for "DNN robustness" which is a much larger scope. The author discussed adversarial robustness and OOD in the introduction and related work sections, which makes it even more confusing when I'm reading the paper. See more in requested changes.

---

> ### Author Response · Authors · 2023-12-01
>
> Thank you for your valuable comments. We detailed answers to your comments below.
>
> > “The title is over-claiming”. “(the) title says the method is for "DNN robustness" which is a much larger scope”.
>
> Thank you for bringing attention to this oversight. In response to this valuable feedback, we have updated the title to “**Sharpness-Aware Minimization Scaled by Outlier Normalization for Improving Robustness on Noisy DNN Accelerators**”. We appreciate the suggestion and believe this title better conveys the essence of our work by narrowing the scope.
>
> > “The author discussed adversarial robustness and OOD in the introduction and related work sections, which makes it even more confusing when I'm reading the paper.”
>
> We have removed all instances of adversarial robustness and OOD in the introduction (**Section 1**) to avoid confusion. Moreover, we moved our experiments on OOD and quantization robustness to **Appendix C** and solely focus on robustness to noisy weights in the main paper to better reflect the scope of our work. Lastly, we added a new paragraph in the related work (**Section 2**) to showcase the differences between adversarial robustness and hardware robustness and to clarify the motivation behind our work. We hope these manuscript changes successfully alleviate any confusion in the updated manuscript. We believe that the discussed changes also address the reviewer’s points 1 and 2 of the high-level requested changes so we will be skipping to point 3.
>
> > “Table 1 & 2 shows the accuracy comparisons between the different methods on CIFAR-10 and CIFAR-100. I think they are just showing the vanilla accuracy, but why the captions say they show the "Generalization performance" as no noise is added during evaluation?”
>
> That is correct - Tables 1 and 2 showcase the vanilla accuracy, which we refer to as generalization performance in the noiseless setting.
>
> > “Could you explain a bit more about how the first two paragraphs in the introduction sections relate to the main contribution of this paper? I don't quite understand why the energy consumption and in-memory computation for DNNs can be a motivation of this robustness study”
>
> In this work, we focus on the class of noise that arises from unreliable hardware accelerators such as the memristor which we study in the paper. These memristors are an emerging technology allowing in-memory computing, promising important energy consumption reduction. With the increase in size and energy consumption of recent DNNs, adopting highly energy-efficient technologies is extremely important and can only be brought to fruition if adequate methods are developed to improve robustness to hardware noise at inference time. Our method, SAMSON, constitutes one of such methods. We have updated the text in **Section 1** to better convey this message.
>
> > “if you want to claim the traditional adversarial training [1] is inefficient, you might want to cite it properly instead of only talking about general DNN accelerator design without the context of adversarial training. Besides, would be great to also mention the efficient versions of adversarial training, e.g., [2], and discuss their limitations that your method can address.”
>
> We would like to clarify that when we talk about efficiency in our paper, it is related to efficient hardware designs that improve the energy consumption of DNNs at inference time. This is not related to improving the efficiency of sharpness-aware minimization methods, as it is the focus of the suggested reference [2].
>
> > “In the second paragraph in the introduction section, it says "However, such approaches typically rely on noise simulations from the target hardware to which the DNN will be deployed.". I'm confused about the definition of the "noise simulations from the target hardware", probably you can move its definition from related work to introduction.”
>
> Thank you for your suggestion. We have updated the text at the end of **paragraph 2** of **Section 1** to clarify these notions.
>
> > “Please provide citations to "existing methods" when you say "Moreover, existing robustness methods provide a trade-off between DNN performance and DNN robustness, decreasing the former to increase the latter." in the introduction section.
> ”
>
> We moved this sentence closer to the appropriate citations and introduced the word “these” to avoid ambiguity.
>
> > “If adversarial robustness is the scope, please discuss related works on SAM+adversarial robustness evaluation, e.g., [3]. If not, please clarify this point and explain why Sun et al. (2021)[4]'s work on adversarial robustness is the most related.”
>
> Adversarial robustness is not the scope, but hardware robustness is. We believe that our changes throughout **Sections 1 and 2** now clearly reflect this.

---

> > ### Author Response · Authors · 2023-12-01
> > **(continuation)**
> >
> > > “In the last paragraph in the introduction section, you say "By promoting sharpness adaptivity based on the outlier weights, we show that SAMSON’s sharpness measure has a high correlation with model robustness. In other words, SAMSON’s objective may be used during training in combination with existing robustness techniques to increase DNN robustness at inference time.". However, I don't think the two sentences have the same meaning, right?”
> >
> > We have updated the **last paragraph** in **Section 1** to better reflect the connections between these two sentences. Thank you for your suggestions on improving the clarity of our revised manuscript.

---

### Review · Reviewer_biRM · 2023-11-16

**Summary Of Contributions:**

This article presents a variant to the Adaptive SAM algorithm, which they named SAMSON, by adding the $\parallel w\parallel_p$ term to the restriction applied to $\epsilon$. The analytical $\epsilon^*$ is obtained to make the objective easy to compute. This article then conducts experiments for classic architectures on standard image benchmarks, and shows several findings including: (1) similar or marginally better generalization performance, (2) better robustness in the setting of noisy weights due to conductance variation, and (3) better robustness for OOD sampes and post-training quantization.

**Audience:**

Yes

**Broader Impact Concerns:**

None.

**Claims And Evidence:**

Yes

**Requested Changes:**

- Make theoretical justification of the proposed objective and neighbourhood definition. In a rigorous and principled way, explain why this objective can address the sharpness issue that ASAM could not solve.
- Make a clear boarder of the proposed method: which types of sharpness does it address, and which is outside the scope of this article.
- Make theoretical justification on the $p$-norm term. Explain which $p$ is suitable for which types of tasks or loss functions from a theoretical point of view.
- Have a more detailed explanation to Fig 1 by explaining how these lines are generated and the meaning of different colors.
- Present additional experiments in the noisy weight setting and show the proposed method has a significant improvement over baseline methods.

**Strengths And Weaknesses:**

Strengths:
- This article is very well motivated from the sharpness perspective. It is a very important aspect of neural networks that is related to many other properties including robustness and generalization. The background of this area and its relationship to the other properties are clearly and thoroughly presented in the first three sections.
- This article has very extensive experiments across a variety of classic neural network architectures and different datasets. The generalization results indicate the proposed method can achieve similar generalization performance as other methods. The setting of noisy weights caused by conductance variation is a potentially useful use case in practice.

Weakness:
- The article lacks justification for the proposed objective. Despite there is a high-level intuition that adding the p-norm term can address the sharpness issue in a better way, it is unclear why and how this is the case from the theoretical perspective. It is not discussed what types of sharpness can be addressed by the proposed method and what cannot. It is also not discussed which $p$ is suitable for different scenarios; the article simply regards it as a hyperparameter to tune, without a principled way to determine it.
- Although the experimental results for robustness under noisy weights show improvement over ASAM and SAM, the results do not demonstrate a significant enhancement, which limits the contribution of this article.

---

> ### Author Response · Authors · 2023-12-01
>
> Thank you for your valuable comments. We address your feedback below.
>
> > “Make theoretical justification of the proposed objective and neighbourhood definition. In a rigorous and principled way, explain why this objective can address the sharpness issue that ASAM could not solve.”
>
> We provide additional discussions between SAMSON and ASAM in **Appendix F**.
>
> > “Make a clear boarder of the proposed method: which types of sharpness does it address, and which is outside the scope of this article.”
>
> We include a discussion on the types of sharpness our method addresses in **Appendix F.1**.
>
> > “Make theoretical justification on the $p$-norm term. Explain which $p$ is suitable for which types of tasks or loss functions from a theoretical point of view.”
>
> We would like to point out that norm terms were treated as hyperparameters in the ablations conducted by SAM and ASAM papers. Even though such norm ablations are related to the effect of the fixed (non-adaptive) neighborhood regions of all weights, and our method uses different norms to control the importance of the outlier weights in the adaptive neighborhood region of each weight, we believe the same hyperparameter concept applies here. Since both $p=2$ and $p=\infty$ outperform one another depending on the setting, we recommended treating our $p$-norm term as a hyperparameter in practice.
>
> > “Have a more detailed explanation to Fig 1 by explaining how these lines are generated and the meaning of different colors.”
>
> Thank you for this suggestion. We updated the **second-to-last paragraph** of **Section 4** with these explanations in the revised manuscript.
>
> > “Present additional experiments in the noisy weight setting and show the proposed method has a significant improvement over baseline methods.”
>
> We included an additional experiment in the noisy weight setting using a large-scale Transformer model for machine translation. The results are presented in **Figure 3** of **Section 6.2.1**. Overall, we observe that SAMSON$_\infty$ promotes the best robustness across all noise regimes even without applying weight clipping (left figure). When weight clipping is applied, our method variants are the most viable solution to achieve higher robustness in low to mid-noise regimes. Moreover, we see that both of our method variants exhibit the best generalization performance in the noiseless setting both with and without weight clipping.

---

### Author Response · Authors · 2023-12-01
**General response**

We would like to thank the reviewers for their time and valuable comments. We are happy they found our paper to be well written (**Reviewer V41i**) and easy to follow (**Reviewer V41i**), with our method being well motivated (**Reviewers V41i, biRM**) and our experimental settings being very extensive (**Reviewer biRM**).

To address specific feedback from the reviewers, we made the following changes to the manuscript (highlighted in blue):
- We changed the title to **“Sharpness-Aware Minimization Scaled by Outlier Normalization for Improving Robustness on Noisy DNN Accelerators”** to address **Reviewer GL5H**’s concerns about the previous title having a wide scope.
- We moved our additional robustness experiments on out-of-distribution generalization and quantization to **Appendix C**, following **Reviewer GL5H**’s concerns about the clarity of the main scope of the work. We also updated **Sections 1 and 2** to better reflect our scope and contributions.
- We added **Section 5.1** which presents generalization performance experiments with large-scale Transformer models and additional tasks (**Table 4**), as suggested by **Reviewer V41i**.
- We added **Section 6.2.1** which introduces additional experiments in the noisy weight setting (**Figure 3**), as suggested by **Reviewer biRM**.
- We added **Appendix E.1** which presents the Pearson correlations between sharpness and robustness (**Table 19**), as requested by **Reviewer V41i**.
- We added **Appendix F** which addresses further discussions and limitations of our method, as suggested by **Reviewer biRM**.

We address each reviewer individually and provide detailed responses to their comments. We would like to thank the reviewers once again for their valuable feedback.

---

### Decision · Action_Editor_kWFK · 2024-01-03

**Recommendation:** Reject

**Comment:**

This paper investigates techniques to enhance the robustness of DNN model to hardware noise when deployed on analog-based, in-memory computing accelerators. The proposed method builds on existing technique, specifically adaptive Sharpness-Aware Minimization, by incorporating the Lp-norm of the weight tensor into the loss sharpness regularization term to promote a smoother loss landscape based on the range of weight distribution.  The method is evaluation on several popular CNN models and CIFAR/ImageNet datasets.

The paper addresses an important issue in improving the performance and robustness of DNN models when deployed on analog-based, noisy hardware. The proposed method is intuitive and well-motivated, although only a simple extension of existing method.

The reviewers have raised several concerns:

1). The efficacy of the proposed methods is not thoroughly justified, as the performance and robustness improvement is only marginal over existing techniques, and the paper lacks any theoretical analysis or guarantees.

2). The paper lacks clarification in addressing the problem to be solved and could be better organized. For example, the paper claims to provide general improvement on model robustness, but failed to provide adequate justification. During rebuttal, the authors partially addressed the concerns by limiting their focus to only hard-ware noise robustness problem.

3). The evaluation is limited to relatively outdated models. During rebuttal, the authors provide some more data on vision transformer models.

In summary, the paper tackles a significant problem, and the proposed method is reasonably well-motivated and explained. However, the results do not provide strong evidence to support the approach, and the claims made are somewhat confusing. One reviewer recommends acceptance, while two reviewers lean towards rejection. During the rebuttal process, the authors provided additional data and clarifications, but they did not fully address the concerns. Therefore, I am inclined towards a weak rejection.

**Audience:**

The paper tackles an important problem in the deploying DNN models in analog-based, in-memory computing accelerators. The idea is reasonably interesting. However, the experimental results donot provide clear evidence of the method’s efficacy, making it difficult to assess its impact.

**Claims And Evidence:**

The proposed method represents a simple extension of existing techniques and its details are clearly presented. However, the experimental results do not demonstrate a significant improvement over existing methods. In addition, reviewers have pointed out some confusion regarding the scope of the methods claimed by the paper to be generalizable to DNN robustness. In response to these concerns, the authors have modified their claims during the rebuttal process.